# TreeFinder: A US-Scale Benchmark Dataset for Individual Tree Mortality Monitoring Using High-Resolution Aerial Imagery

**Zhihao Wang**[1], **Cooper Li**[1], **Ruichen Wang**[1], **Lei Ma**[1]
**George Hurtt**[1], **Xiaowei Jia**[2], **Gengchen Mai**[3], **Zhili Li**[1], **Yiqun Xie**[1*]
[1]University of Maryland, [2]University of Pittsburgh, [3]University of Texas at Austin
{zhwang1, cligeog25, ruichenw, lma6, gchurtt, lizhili, xie}@umd.edu,
xiaowei@pitt.edu, gengchen.mai@austin.utexas.edu

## Abstract

Monitoring individual tree mortality at scale has been found to be crucial for understanding forest loss, ecosystem resilience, carbon fluxes, and climate-induced impacts. However, the fine-granularity monitoring faces major challenges on both the data and methodology sides because: (1) finding isolated individual-level tree deaths requires high-resolution remote sensing images with broad coverage, and (2) compared to regular geo-objects (e.g., buildings), dead trees often exhibit weaker contrast and high variability across tree types, landscapes and ecosystems. Existing datasets on tree mortality primarily rely on moderate-resolution satellite imagery (e.g., 30m resolution), which aims to detect large-patch wipeouts but is unable to recognize individual-level tree mortality events. Several efforts have explored alternatives via very-high-resolution drone imagery. However, drone images are highly expensive and can only be collected at local scales, which are therefore not suitable for national-scale applications and beyond. To bridge the gaps, we introduce TreeFinder, the first high-resolution remote sensing benchmark dataset designed for individual-level tree mortality mapping across the Contiguous United States (CONUS). Specifically, the dataset uses NAIP imagery at 0.6m resolution that provides wall-to-wall coverage of the entire CONUS. TreeFinder contains images with pixel-level labels generated via extensive manual annotation that covers forested areas in 48 states with over 23,000 hectares. All annotations are rigorously validated using multi-temporal NAIP images and auxiliary vegetation indices from remote sensing imagery. Moreover, TreeFinder includes multiple evaluation scenarios to test the models' ability in generalizing across different geographic regions, climate zones, and forests with different plant function types. Finally, we develop benchmarks using a suite of semantic segmentation models, including both convolutional architectures and more recent foundation models based on vision transformers for general and remote sensing images. Our dataset and code are publicly available on Kaggle and GitHub: https://www.kaggle.com/datasets/zhihaow/tree-finder and https://github.com/zhwang0/treefinder.

## 1 Introduction

Forests play a critical role in the ecological balance of the Earth, significantly influencing global carbon cycles [20, 31], biodiversity conservation [8, 7], climate regulation [35, 29], and water

---

*Corresponding author.

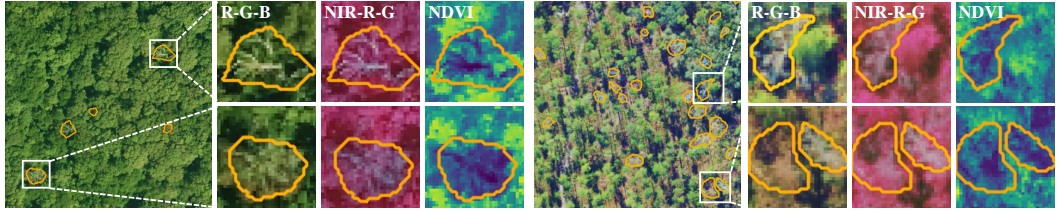

|  (a) An example of uniform forest in Illinois. | (b) An example of complex forest in Florida. |

Figure 1: Examples of scattered tree mortality visualized with three spectral representations: (1) true-color (R-G-B), (2) false-color (NIR-R-G), commonly used in remote sensing to highlight vegetation in red, and (3) vegetation index (NDVI), where brighter colors indicate higher vegetation activities.

resources [9, 18]. The health and stability of forests are increasingly threatened by widespread tree mortality, which can substantially alter carbon storage, disrupt local ecosystems, and increase wildfire risks [47, 14, 32]. While tree wipe-outs as contiguous large patches (e.g. due to wildfire) can be monitored using traditional moderate-resolution remote sensing platforms such as Landsat-8/9 at 30m resolution, scattered (i.e., not contiguous) but widespread tree deaths at the individual level have been largely unmonitored. However, according to recent studies including a *Nature Communications* article [13], such scattered tree deaths have a substantial impact on forest loss and significantly affect carbon budget and sequestration capacity, and serve as critical catalysts for future wildfires. Therefore, accurate mapping and quantification of tree mortality in fine granularity are essential for better ecological monitoring, precise carbon accounting, high-resolution fire risk assessment, and effective forest management policies.

Despite its importance, identifying tree mortality at the individual level remains challenging due to limitations in both data availability and methodological approaches. First, visual signatures of scattered tree deaths are only available in high-resolution images, making traditional remote sensing platforms unsuitable for this detection task. In addition, the high-resolution imagery must have broad-scale geographical coverage (e.g., national level) to answer major carbon cycle questions and inform critical policy and management decisions. Second, compared to traditional geospatial objects or phenomena with sharp contrast and geometric patterns (e.g., buildings), dead trees often present weaker contrast or higher similarity with the surrounding context that makes their pixels harder to separate from the background environment, which may contain varying sunlight, shadows, or landscapes. More importantly, the visual patterns and background contrasts of dead trees can vary significantly across geographic regions due to different climate conditions, forest density, and tree types [3, 33]. This makes it challenging to generalize learned AI models at scale in real-world applications. Finally, the lack of labels that are widely distributed over geographic regions remains a key bottleneck for developing generalizable models for large-scale monitoring.

While efforts have attempted to solve the tree mortality mapping problem, existing datasets are limited in their applicability for monitoring individual-level tree deaths across broad geographical scales. Most current products and monitoring systems are based on low or moderate resolution satellite platforms, such as the 1km-resolution AVHRR imagery [19] or 30m-resolution Landsat imagery [30, 22], which lack sufficient spatial details needed to map fine-granularity tree mortality, including scattered tree deaths. Recent studies have also explored the use of drone imagery at very high resolution (VHR), which is capable of identifying individual tree deaths in localized study sites [43, 27]. However, drone-based monitoring is highly expensive, which significantly constrains its applicability at large scale. Note that even though the labels can be collected at different geographic sites, in practice, it is cost-prohibitive to generate wall-to-wall maps (i.e., spatially contiguous full maps) using these VHR images for applications beyond small local scales. These wall-to-wall maps, though, are often required for scientific research and forest management due to heterogeneity [25].

To address these gaps, we introduce **TreeFinder**, the first high-resolution benchmark dataset designed for individual-level tree mortality mapping across the Contiguous United States (CONUS). Specifically, the dataset uses NAIP imagery at 0.6m-resolution that provides wall-to-wall coverage for the entire CONUS. TreeFinder contains images with pixel-level labels generated via extensive manual annotation that covers forested regions in 48 different states in CONUS with a total area of over 23,000 hectares. The high spatial resolution, combined with broad geographic coverage, offers opportunities to enable accurate identification and delineation of individual tree deaths. Our dead tree annotations

are rigorously validated using multi-temporal data from NAIP imagery, ensuring the accuracy and reliability of labeled dead trees. In addition to the dataset, we further implement a suite of machine learning (ML) methods covering both segmentation models based on traditional convolutional neural networks (CNNs) and more recent foundation models to establish performance benchmarks for individual-level tree mortality monitoring. Finally, to facilitate the evaluation of ML models' generalizability under different scenarios, we associate additional metadata with each image patch to provide information about its geographic location, climate zone, and primary tree type. Considering the large degree of spatial variability, these scenarios are necessary to understand if an ML model is able to maintain robust performance in different conditions, especially those not seen during training. Overall, our TreeFinder dataset and benchmarking initiative not only address major gaps in existing datasets, but also offer opportunities to advance machine learning methods for challenging ecological and environmental science problems at a large scale with cross-region variability. Our open-source dataset and code are available on Kaggle `https://www.kaggle.com/datasets/zhihaow/tree-finder` and GitHub `https://github.com/zhwang0/treefinder`. Our key contributions are summarized as follows:

- We create a large-scale, high-resolution dataset covering 1,000 sites over 48 states in CONUS, with a total area of 23,000 hectares. The 0.6m high-resolution NAIP images at each site are manually annotated for dead trees at the pixel level using both visual features from single NAIP images and temporal differences between multi-temporal NAIP images.

- We develop performance benchmarks using a suite of ML models, including traditional CNN-based segmentation methods and more recent foundation models for general and remote sensing images.

- We integrate metadata on geographic locations, climate zones, and primary tree types to each image patch to enable performance evaluation and model comparison under different scenarios and the spatial variability challenge.

## 2   Related Work

**Existing benchmark datasets on remote sensing semantic segmentation.**    Semantic segmentation has gained increasing attention in the remote sensing domain, as large-scale, pixel-level classification from satellite or aerial imagery provides important and detailed information for the monitoring of diverse Earth surface conditions such as land cover types [37, 44], urban infrastructure [16, 24], and crop growth [45, 5]. Several datasets have been developed for this purpose, including DeepGlobe for land cover segmentation [15], CropHarvest for global crop type mapping using both optical and SAR satellite imagery [45], LoveDA for domain-adaptive segmentation across urban and rural scenes [50], and SAMRS leveraging SAM and existing datasets [48]. These datasets typically focus on well-structured geospatial objects such as buildings, roads, and crop fields, which exhibit strong spatial regularity and clear boundaries. Segmentation of geospatial objects with lower contrast (e.g., individual-level tree deaths) on a large national scale using high-resolution images has been underexplored in existing datasets, as well as generalization across different climate zones and ecological conditions. Our experiments in Sec. 4.2 show that such tasks indeed remain challenging for current segmentation models.

**Existing datasets on tree mortality monitoring.**    Existing datasets remain limited along several critical dimensions, including spatial resolution, geographic coverage, and label availability. While **drone-based datasets** offer very high spatial resolution (e.g., Almorox Crown Dataset [2], FOR-instance [36]), their spatial coverage is highly constrained due to high operational cost, often restricted to localized study areas (e.g., tens or a few hundred hectares). A recent work, *deadtrees.earth* [34], is an encouraging platform effort aiming to support collaborations for tree mortality label collection. However, the dataset relies on drone imagery, which offers centimeter-level resolution but is limited to sample sites due to high cost, constraining its practical applicability for large-scale monitoring tasks. Moreover, according to the paper, its images are biased toward forests located near human settlements as they were originally collected for other purposes. As a result, the data may not be representative of forest ecosystems. On the other hand, coarse-to-moderate resolution imagery from **satellite platforms** makes large-scale coverage possible [40, 41, 38], but the resolution only supports detecting dead trees that form large and contiguous patches and the images lack necessary spatial details to capture fine-granularity tree mortality patterns. Finally, aerial images offer new opportunities to consider both the geographic coverage and resolution [1, 28]. For example, the National Agriculture Imagery

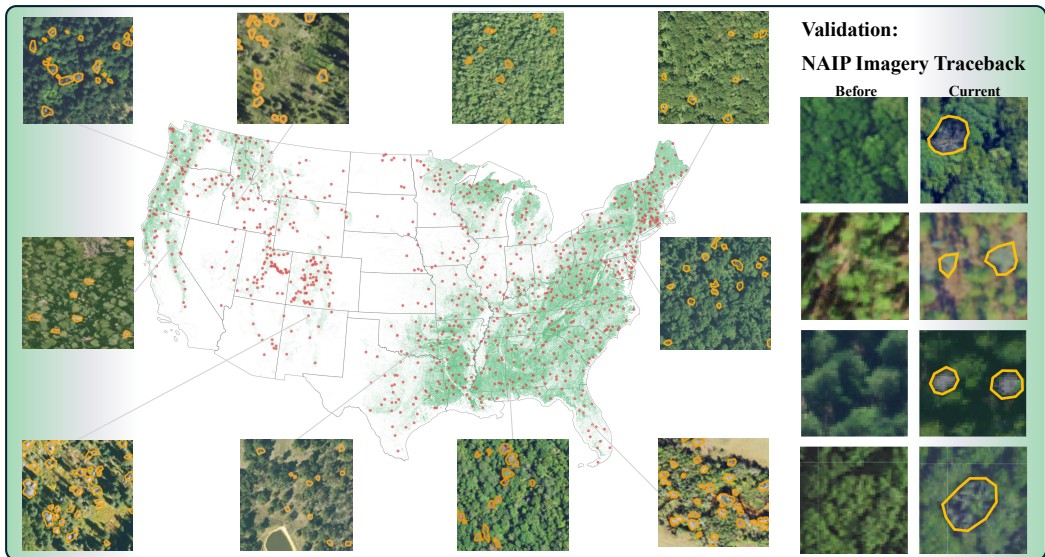

Figure 2: Left: Distribution of the 1000 sites from the Contiguous US and example visualizations of labeled tree deaths. Right: Illustrations of the validation process using multi-temporal images.

Program (NAIP) provides 0.6m resolution images over the entire CONUS region. However, existing datasets have only considered local areas (e.g., the Sierra National Forest in California). In addition, the samples are predominantly generated by ML models, and the manual labels are very limited and not publicly shared. Finally, from the ML model evaluation and benchmarking perspective, existing datasets lack ecological, climate, and geographic diversity, limiting their generalizability across forest types, climatic conditions and locations. As a result, they are not practically suitable for developing models for large-scale monitoring, e.g., national scales.

**ML methods for satellite-based segmentation.** Deep learning has driven major advances in semantic segmentation. CNN architectures such as U-Net [39], DeepLabV3+ [10], and HRNet [49] are widely adopted in satellite-based segmentation tasks thanks to their stable performance. Recent advances in vision transformers (ViTs) have enabled more flexible and scalable modeling of long-range dependencies. Models such as SegFormer [52] introduced hierarchical transformer encoders with efficient multi-scale fusion, while Mask2Former [11] and Segmenter [42] extend transformers to class-agnostic and class-aware segmentation frameworks. In the geospatial domain, foundation models like NASA-IBM Prithvi [26], SpectralGPT [23] and DOFA [53] are pretrained on large-scale satellite datasets and have demonstrated competitive performance. However, many of them are pretrained using specific types of satellite images and the characteristics may not generalize to other platforms. For example, Prithvi and SpectralGPT were pretrained using moderate-resolution multispectral imagery (e.g., Sentinel-2, or Harmonized Landsat and Sentinel-2), where the special designs along the spectral dimensions may not be suitable for platforms with high-resolution and only a few bands (e.g., NAIP). With that said, these developments provide a diverse set of models for our candidate model selection.

## 3 TreeFinder: Dataset Construction

### 3.1 Data Collection, Annotation, and Validation

TreeFinder aims to provide a CONUS-scale, publicly available, and ML-ready dataset using NAIP imagery to support the development of ML-based segmentation methods that have strong generalizability over geographic regions and beneficial for fine-granularity geo-event monitoring at large scale. Specifically, NAIP provides 4-band aerial images including RGB and near-infrared (NIR) channels, with a contiguous coverage over the entire CONUS area. As NAIP has undergone continued enhancements over time, it has historically produced imagery at varying spatial resolutions, including 2 m, 1 m, 0.6 m, and 0.3 m. The varying resolutions and different acquisition conditions (e.g., dates,

viewing angles, sunlight angles) make it challenging to construct a consistent multi-temporal dataset. Thus, we only use the most recent high-resolution imagery collected after 2021, ensuring sufficient spatial granularity required for high-quality, individual-level tree mortality mapping. A small subset of post-2021 imagery is available at 0.3 m in a few regions, and we resampled these images to 0.6 m to maintain a consistent resolution across all samples.

As the scope of TreeFinder focuses on forested areas, where individual-level tree mortality has been shown to have substantial impact on forest health, carbon cycles and wildfire risks, we first use a well-established, national-scale forest cover basemap [21] to define the geographic mask for the sampling. From this mask, we randomly sample sites across the CONUS to generate manually annotated labels for the dead trees. Specifically, we delineate the individual dead trees using polygon-based tools through Google Earth Engine, a cloud-based platform hosting the full NAIP imagery archive. One challenge during the labeling process is that the individual dead trees could have similarity to bare ground when there is weak visual contrast, especially in cases of isolated, stand-alone mortality or when shadows from neighboring trees partially obscure the crown. To mitigate this, we utilize several strategies to confirm the class belongings as shown in Fig. 1: (1) We use multiple spectral representations including the true-color composition using the default visible R-G-B channels, the false-color composition with NIR-R-G channels that are commonly used in remote sensing to highlight vegetation distributions, and indices such as the Normalized Difference Vegetation Index (NDVI) to better observe vegetation activities. In false-color composition, vegetation is shown as red, and in NDVI maps, pixels with brighter colors indicate higher vegetation activities. A tree is delineated and labeled as dead if its crown is structurally distinct from neighboring trees and exhibits complete canopy de-saturation (e.g., a consistent gray or brown color with no visible greenness) in different multi-spectral visualizations. Moreover, we also validate the labels using multi-temporal combinations of historical NAIP images. While the images can have a lower spatial resolution (1m) in earlier years, they can still provide some visual cues (e.g. color shifts or structural changes) to inform the annotations. Finally, the annotations undergo a cross-annotator assessment on a randomly selected 20% subset of the sites, achieving an agreement rate of over 95%. More details on annotations and validations are available in the appendix. In total, we annotate NAIP images at 1,000 unique sites across CONUS as shown in Fig. 2.

To facilitate the evaluation of ML models' generalizability under diverse conditions (detailed in Sec. 3.3), we enrich each labeled NAIP image with metadata on **geographic locations**, **climate conditions** and **primary tree types**. Specifically, we include information about latitude, longitude, and state for each image to indicate its location. We also assign a Köppen–Geiger climate classification label to each image using the latest gridded global product, which captures present climatic regimes based on temperature and precipitation seasonality [4]. For tree type information, we overlay each NAIP image with the Individual Tree Species Parameter Maps from the U.S. Department of Agriculture's Forest Service [46], which provide estimates of different tree type composition across forested areas in the U.S. Each image is assigned its primary tree type based on the most frequent tree type. The primary type is often used to reduce uncertainty in tree species mapping. Here we did not include the detailed proportions of tree types for the same reason, and also because we are only using it for later generalization tests across primary tree types.

## 3.2   ML-Ready Dataset Preparation

In this paper, the ML-ready dataset means that the data is preprocessed into standard input and output formats for convenient use of ML model training and evaluation. Specifically, the raw NAIP scenes and annotated polygons are converted to fixed-size, model-compatible image patches with consistent spatial dimensions. Specifically, each annotated image is split into non-overlapping patches of $224 \times 224$ pixels. Based on our visual inspection, a patch size of $224 \times 224$ is sufficiently large to capture full details that are needed to identify individual-level dead trees. This leads to a total of $N = 15,489$ image patches, and the input images form a tensor $\mathbf{X} \in \mathbb{R}^{N \times 224 \times 224 \times 4}$ with 4 bands in each image, and the output labels form a tensor $\mathbf{Y} \in \{0,1\}^{N \times 224 \times 224}$, where 1 indicates a pixel of a dead tree. Finally, since aerial images do not always align with the orthogonal directions of the geographic reference systems and their shapes may change after projection and ortho-rectification, there are often areas with empty values in an image patch. Thus, we also provide a binary masking layer $\mathbf{M} \in \{0,1\}^{N \times 224 \times 224}$ to help exclude the null pixels during evaluation. We provide the dataset in multiple formats for convenience. First, we provide the original GeoTIFF format, which preserves all the spatial referencing information (e.g., geographic coordinate system, projection) of each image

for visualization or further integration with other spatial data. Second, we develop and share a Python library to load, filter, and batch the dataset based on user-defined criteria. This can convert the data into other more direct formats for ML models: (1) Numpy array format, where the tensors are stored as .npy files; and (2) TFRecords format, which supports easier usage with ML models.

## 3.3 Scenarios for Generalization Test

To more comprehensively assess the generalizability of ML models under diverse geographical, climatic, and ecological conditions, we define four benchmarking scenarios: (1) baseline (easiest) scenario with random sampling, (2) generalization over geographic regions, (3) generalization over climate zones, and (4) generalization over different types of forests. There are certain connections between generalization over geographic regions and generalization over climates and forest types, as different climates or forest types are also in different regions. We include geographic generalization as a separate scenario as it can be considered as an integration of many factors (e.g., different regional management practices on forest recovery efforts), and it is common in practice to have labels highly localized in certain regions. Each scenario represents a practical deployment situation, and these are important to understand for applications, as the labeled set often only covers a small fraction of the entire study area (e.g., CONUS) in large-scale monitoring tasks.

- **Random split with incremental training sizes.** In this baseline scenario, we evaluate the overall model performance using a standard random split, where 20% of the labeled patches are held out as a fixed test set. The remaining 80% of the dataset is used for training and validation, in which we denote the training data ratio as $\alpha$. To examine the influence of data size on model performance, we subsample the training set as incremental proportions by varying $\alpha$ from 10% to 80%, while keeping the test set fixed. The validation set is randomly sampled as 10% of the training set. While the baseline scenario is the easiest among the four scenarios, the additional evaluations with different training data proportions reflect real-world settings where labeled data may be limited and help understand the model's sensitivity to sample size.

- **Cross-region scenarios.** To test the spatial generalizability of different ML models on our benchmark dataset, we consider the following splits: (1) Western-eastern split: The dataset is divided into western and eastern regions of the CONUS, using the Mississippi River as a natural boundary. Models are trained in one region and evaluated on the other. As explained earlier, the variability across locations can be considered as an aggregation of factors including climates, forest types, and others. (2) One state vs. all: This is a challenging scenario where the training samples come from one single state and the evaluation is performed on all remaining states. To set up a concrete example, we use Colorado as the single state for training, as the state is well-known for its forested mountains, and the tree mortality problems have been widely observed and studied in the area [51, 6]. This setup also reflects practical scenarios where certain states start the monitoring programs earlier on tree mortality events and thus contribute disproportionately to the training data than others.

- **Cross-climate scenarios.** These scenarios evaluate model generalization across climate regimes by training on data from one set of climate conditions and then test it on the others. First, we build one group of climate zones that include Mediterranean, humid subtropical zone, humid continental zone, etc., comprising approximately 50% of our dataset, and the remaining, such as the humid subtropical climate zone, as the other group. Second, we consider a more challenging scenario, where the training is performed on a climate zone with significantly smaller number of samples and then tested on the rest of the zones. Specifically, we use the humid continental zone as the training climate zone and the rest for testing.

- **Cross-forest-type scenarios.** Finally, we design scenarios to test model generalization across different primary forest types. First, we construct a relatively easier case with a broad training set consisting of the top five most frequent primary tree types–maple, pine, oak, Douglas fir, and cottonwood–which together account for approximately 50% of all labeled samples. Models are trained on this subset and evaluated on all other primary forest types. Second, we define a more challenging generalization scenario, in which the model is trained using only one primary tree type, maple, and evaluated on all others.

**Metrics.** Performance is evaluated using standard segmentation metrics, including precision, recall, F1 score, intersection-over-union (IoU), and overall accuracy. The segmentation statistics across all

image patches are first aggregated together and then used to compute the final metrics, rather than averaging per-patch values. Except for accuracy, the other metrics need to be calculated per class to better reflect the model's performance. Thus, we include both the metrics for the target class (dead trees) and for both (average of the target class and background class) in the result tables.

## 4 Experiments

### 4.1 Candidate methods

To benchmark model performance, we consider a set of segmentation architectures covering both CNN-based and transformer-based designs. This selection captures convolutional methods focusing on localized feature extraction as well as more recent transformer-based foundation models pretrained using both general images and remote sensing images. Specifically, we consider the following candidate models:

- **U-Net**: An encoder–decoder network with skip connections. Our U-Net is trained from scratch to provide a baseline with localized spatial modeling and no reliance on pretraining [39].

- **DeepLabV3+**: A CNN-based model with a ResNet-50 backbone, leveraging atrous spatial pyramid pooling to aggregate multi-scale contextual features. The model was pretrained on ImageNet [10] and we customized it with input and output modifications.

- **Vision Transformer (ViT)**: A patch-based transformer with a lightweight transposed convolution decoder that upsamples hidden features back to full resolution [17]. The model is pretrained on ImageNet. Following common strategies, we added a segmentation head to customize it for semantic segmentation [52].

- **SegFormer**: A hierarchical transformer architecture that is designed for semantic segmentation [52]. It uses multi-scale feature encoding with a lightweight decoder, enabling better spatial representation and hierarchical feature extraction. SegFormer was pretrained on pretrained on ADE20k.

- **Mask2Former**: A transformer-based framework that combines a Swin-Tiny backbone with multi-scale deformable attention and a class-agnostic mask prediction head [12]. It models segmentation as a set prediction task using masked attention, and we used the pretrained weights on ADE20K.

- **DOFA**: A multimodal foundation model specifically designed for remote sensing images [53]. It uses wavelengths to embed different spectral bands into a unified feature space, enabling the learning of shared representations across channels. DOFA is pretrained on multi-sensor remote sensing imagery, including Sentinel 1/2 and NAIP. As it is not limited to specific remote sensing sensors and considers NAIP in pretraining, we included it as part of the evaluation. We did not include Prithvi and SpectralGPT as they are specifically designed for multispectral images (e.g., 10+ bands) and pretrained using moderate-resolution remote sensing images, which are largely distinct from NAIP.

All models are trained using the training set of TreeFinder (varying by evaluation scenarios), or fine-tuned if pretrained weights are available. We use a batch size of 32, an initial learning rate of $e^{-4}$ with the Adam optimizer. All models are trained for up to 100 epochs using a combined loss function from binary cross-entropy loss and dice loss to mitigate class imbalance issues. We also applied early stopping based on validation loss to prevent overfitting. More details on training are available in the appendix.

### 4.2 Results

**Random split performance and impact of training size.** Table 1 shows model performance under a standard 80-20 random train-test split, using 10% of the training samples for validation. Among all models, it is interesting to see that Mask2Former achieves the highest F1, precision, and IoU, while U-Net has the highest recall. The differences between U-Net, DeepLabV3+, and SegFormer are within 2-3% in this baseline scenario. DOFA did not perform well on the metrics compared to the others, potentially due to the trade-off between its goal to cover broader sensing platforms with different sets of spectral bands, and the performance on specific types of platforms. The accuracy for all models remains very high because the problem has imbalanced class distribution where dead trees

Table 1: Results for the random split scenario (numbers shown as %), with standard deviations across three runs. Best results are bolded.

| Model | F1 | | Precision | | Recall | | IoU | | Accuracy |
| --- | --- | --- | --- | --- | --- | --- | --- | --- | --- |
| | Dead Tree | All | Dead Tree | All | Dead Tree | All | Dead Tree | All | All |
| U-Net | 46.1 ± 7.1 | 72.9 ± 3.5 | 35.7 ± 6.8 | 67.8 ± 3.4 | **65.6 ± 5.2** | **82.6 ± 2.6** | 30.1 ± 5.9 | 64.8 ± 2.9 | 99.4 ± 0.0 |
| DeepLabV3+ | 49.7 ± 8.7 | 74.7 ± 4.4 | 49.6 ± 7.1 | 74.7 ± 3.5 | 49.9 ± 10.4 | 74.8 ± 5.2 | 33.4 ± 7.5 | 66.5 ± 3.8 | **99.6 ± 0.0** |
| ViT | 43.6 ± 9.0 | 71.7 ± 4.5 | 45.1 ± 8.2 | 72.4 ± 4.1 | 42.9 ± 11.9 | 71.3 ± 5.9 | 28.2 ± 7.4 | 63.9 ± 3.7 | **99.6 ± 0.0** |
| SegFormer | 47.8 ± 8.3 | 73.8 ± 4.2 | 49.2 ± 5.4 | 74.5 ± 2.7 | 47.0 ± 11.7 | 73.4 ± 5.8 | 31.7 ± 7.4 | 65.7 ± 3.7 | 99.6 ± 0.1 |
| Mask2Former | **51.9 ± 5.5** | **75.8 ± 2.8** | **55.5 ± 8.7** | **77.6 ± 4.3** | 49.4 ± 6.7 | 74.6 ± 3.3 | **35.1 ± 5.1** | **67.4 ± 2.6** | **99.6 ± 0.0** |
| DOFA | 29.2 ± 5.6 | 64.5 ± 2.8 | 31.1 ± 2.9 | 65.4 ± 1.4 | 28.6 ± 9.1 | 64.2 ± 4.5 | 17.2 ± 3.9 | 58.3 ± 1.9 | 99.5 ± 0.1 |

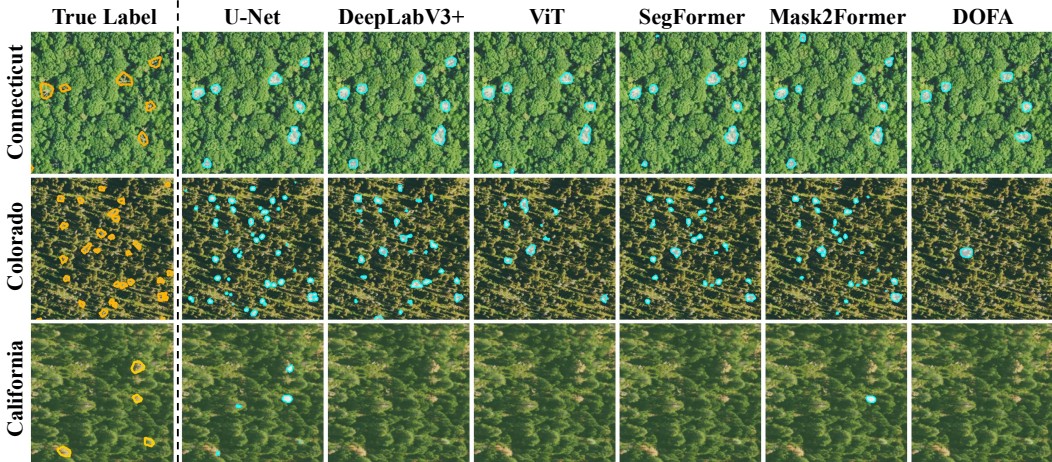

Figure 4: Visualization of example segmentation results from models trained with 80% of the dataset under the random split scenario.

account for a small proportion of the total number of trees. However, their impact on forest health and carbon stock potential is substantial [13]. For example, if a tree dies, not only it will no longer contribute to continued carbon sequestration, but also the existing carbon stock will be taken away, turning to emissions. In Fig. 3, we show the F1 score results by incrementally increasing the training set size from 10% to 80%, while keeping the test set fixed. Results with unstable or poor performance at very small sample size are not included. In general, all models show performance improvements with more training data. DOFA follows the same trend but its overall performance remains lower than the other models. Fig. 4 visualizes several examples of results using models with 80% training data.

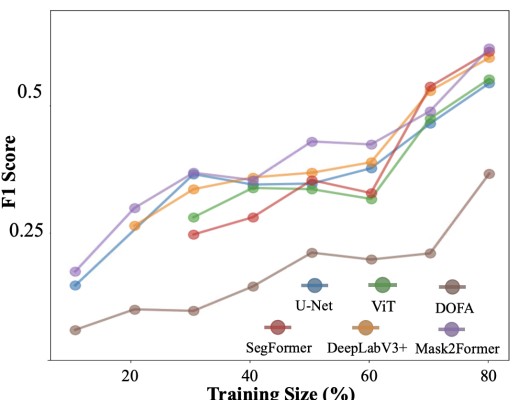

Figure 3: F1 score vs. training data size.

**Cross-region generalizability.** Table 2 presents the results on model generalizability across different spatial regions. Compared to the model performance in the random split setting, the overall performance in this test setting drops significantly. For example, the F1 score has up to 50% relative decrease in the W-E scenario and 70% in the single-state scenario (Colorado - Others), which confirms the increased difficulty of generalizing across geographic domains. Among these three scenarios, SegFormer and Mask2Former take most of the top-ranking positions for F1 score, precision, and IoU. U-Net shows the best performance in recall. It is worth noting that in the single-state scenario, most models such as SegFormer and Mask2Former have higher precision than recall, whereas the pattern is the opposite for U-Net, showing different model tendencies. DOFA and ViT in this case still show relatively lower performance. Comparing the scenarios, W-E shows slightly better results

Table 2: Results for cross-region scenarios: **E - W**: train on eastern and test on western states; **W - E**: train on western and test on eastern states; **CO**: train on Colorado and test on all other states. All values are shown as %, and best values are bolded.

| Model | Scenario | F1 | | Precision | | Recall | | IoU | | Accuracy |
|-------|----------|---------|-----|-----------|-----|--------|-----|---------|-----|----------|
| | | Dead Tree | All | Dead Tree | All | Dead Tree | All | Dead Tree | All | |
| U-Net | E - W | 16.1 | 57.9 | 11.1 | 55.5 | **29.4** | **64.4** | 8.8 | 54.0 | 99.3 |
| DeepLabV3+ | E - W | 20.5 | 60.2 | 20.8 | 60.3 | 20.3 | 60.0 | 11.4 | 55.5 | 99.6 |
| ViT | E - W | 3.8 | 51.4 | 2.1 | 51.0 | 16.9 | 57.5 | 1.9 | 49.9 | 97.9 |
| SegFormer | E - W | **21.8** | **60.8** | 18.1 | 59.0 | 27.3 | 63.5 | **12.2** | **55.9** | 99.5 |
| Mask2Former | E - W | 21.6 | 60.7 | **23.8** | **61.8** | 19.8 | 59.8 | 12.1 | **55.9** | **99.7** |
| DOFA | E - W | 4.8 | 52.0 | 2.8 | 51.3 | 18.4 | 58.4 | 2.4 | 50.3 | 98.2 |
| U-Net | W - E | 13.9 | 56.8 | 40.8 | 70.2 | 8.4 | 54.2 | 7.5 | 53.5 | **99.5** |
| DeepLabV3+ | W - E | 15.0 | 57.4 | **41.2** | **70.4** | 9.1 | 54.5 | 8.1 | 53.8 | **99.5** |
| ViT | W - E | 13.4 | 56.6 | 24.9 | 62.2 | 9.2 | 54.5 | 7.2 | 53.3 | **99.5** |
| SegFormer | W - E | **22.8** | **61.3** | 38.7 | 69.2 | **16.1** | **58.0** | **12.8** | **56.2** | **99.5** |
| Mask2Former | W - E | 21.4 | 60.6 | 38.5 | 69.1 | 14.8 | 57.4 | 12.0 | 55.8 | **99.5** |
| DOFA | W - E | 4.4 | 52.0 | 7.0 | 53.3 | 3.2 | 51.5 | 2.2 | 50.8 | 99.4 |
| U-Net | CO | 10.2 | 54.4 | 5.8 | 52.8 | **40.1** | **68.8** | 5.4 | 51.3 | 97.3 |
| DeepLabV3+ | CO | 15.4 | 57.6 | 20.2 | 59.9 | 12.5 | 56.2 | 8.4 | 53.9 | 99.5 |
| ViT | CO | 6.3 | 53.0 | 13.1 | 56.4 | 4.1 | 52.0 | 3.2 | 51.4 | 99.5 |
| SegFormer | CO | **17.5** | **58.7** | 27.8 | 63.7 | 12.8 | 56.3 | **9.6** | **54.6** | 99.5 |
| Mask2Former | CO | 11.6 | 55.7 | **29.7** | **64.7** | 7.2 | 53.6 | 6.1 | 52.9 | **99.6** |
| DOFA | CO | 10.2 | 54.9 | 10.2 | 55.0 | 10.2 | 54.9 | 5.4 | 52.4 | 99.3 |

Table 3: Performance across shifted domains in climate zones and primary tree types. All values are percentages %, and best values are bolded.

| Model | Scenario | F1 | | Precision | | Recall | | IoU | | Accuracy |
|-------|----------|---------|-----|-----------|-----|--------|-----|---------|-----|----------|
| | | Dead Tree | All | Dead Tree | All | Dead Tree | All | Dead Tree | All | |
| U-Net | Climate | 18.6 | 59.1 | 17.3 | 58.5 | 20.1 | 59.8 | 10.3 | 54.8 | 99.3 |
| DeepLabV3+ | Climate | 25.8 | 62.8 | 31.4 | 65.5 | 21.9 | 60.9 | 14.8 | 57.2 | **99.5** |
| ViT | Climate | 21.6 | 60.7 | 23.4 | 61.5 | 20.1 | 59.9 | 12.1 | 55.8 | 99.4 |
| SegFormer | Climate | 28.0 | 63.9 | **33.7** | **66.7** | 24.0 | 61.9 | 16.3 | 57.9 | **99.5** |
| Mask2Former | Climate | **29.8** | **64.7** | 25.0 | 62.4 | **36.8** | **68.2** | **17.5** | **58.4** | 99.3 |
| DOFA | Climate | 12.9 | 56.3 | 15.7 | 57.6 | 10.9 | 55.3 | 6.9 | 53.1 | 99.4 |
| U-Net | Climate-hard | 19.9 | 59.8 | 26.2 | 62.9 | 16.1 | 58.0 | 11.1 | 55.3 | 99.5 |
| DeepLabV3+ | Climate-hard | 18.0 | 58.9 | 36.2 | 67.9 | 12.0 | 56.0 | 9.9 | 54.7 | **99.6** |
| ViT | Climate-hard | 11.4 | 55.6 | 22.7 | 61.2 | 7.6 | 53.7 | 6.0 | 52.8 | 99.5 |
| SegFormer | Climate-hard | **22.6** | **61.2** | 33.1 | 66.4 | **17.1** | **58.5** | **12.7** | **56.1** | 99.5 |
| Mask2Former | Climate-hard | 15.4 | 57.6 | **45.8** | **72.7** | 9.2 | 54.6 | 8.3 | 54.0 | **99.6** |
| DOFA | Climate-hard | 4.7 | 52.1 | 3.9 | 51.8 | 6.0 | 52.7 | 2.4 | 50.7 | 99.0 |
| U-Net | Forest | 35.3 | 67.5 | 34.3 | 67.0 | 36.3 | 68.0 | 21.4 | 60.4 | 99.5 |
| DeepLabV3+ | Forest | 36.1 | 67.9 | 36.9 | 68.3 | 35.3 | 67.5 | 22.0 | 60.8 | 99.5 |
| ViT | Forest | 26.5 | 63.1 | 27.3 | 63.5 | 25.8 | 62.8 | 15.3 | 57.4 | 99.4 |
| SegFormer | Forest | **40.1** | **69.9** | **41.5** | **70.7** | 38.7 | 69.2 | **25.1** | **62.3** | **99.5** |
| Mask2Former | Forest | 38.6 | 69.2 | 36.8 | 68.3 | **40.6** | **70.1** | 23.9 | 61.7 | **99.5** |
| DOFA | Forest | 13.9 | 56.8 | 14.4 | 57.0 | 13.3 | 56.5 | 7.5 | 53.4 | 99.4 |
| U-Net | Forest-hard | 14.7 | 57.0 | 9.6 | 54.7 | 31.5 | 65.3 | 7.9 | 53.3 | 98.8 |
| DeepLabV3+ | Forest-hard | 21.5 | 60.7 | 27.5 | 63.6 | 17.7 | 58.8 | 12.1 | 55.8 | **99.6** |
| ViT | Forest-hard | 19.9 | 59.7 | 14.5 | 57.1 | 31.7 | 65.5 | 11.1 | 55.1 | 99.1 |
| SegFormer | Forest-hard | **37.7** | **68.7** | **38.3** | **69.0** | 37.1 | 68.5 | **23.2** | **61.4** | **99.6** |
| Mask2Former | Forest-hard | 11.1 | 55.4 | 10.4 | 55.0 | 11.9 | 55.8 | 5.9 | 52.6 | 99.4 |
| DOFA | Forest-hard | 14.3 | 56.9 | 11.2 | 55.4 | 20.0 | 59.7 | 7.7 | 53.4 | 99.1 |

compared to E-W on average. The reason could be that the west side covers more conditions (e.g., local climates) or the task there is more challenging with less contrast to the background landscape. The single state case shows significantly reduced scores due to the lack of sufficient representative samples.

**Cross-climate and cross-forest-type generalizability.** The results for cross-climate and cross-forest-type generalization are shown in Table 3. We skipped the results for train-test group swaps (i.e., similarly like the 2nd row "W-E" in Table 2) due to the space limit, and the full tables are available in the appendix. In Table 3, "Climate" and "Forest" represent the relatively easier scenarios where about half of the data from certain climate zones or forest types are used for training and the rest

for testing. The "-hard" modes are the cases where only one climate zone or forest type is used as training, as described in Sec. 3.3. The general trend is similar as before, where major decreases in scores are observed compared to the random-split case due to the intrinsically higher difficulty. Overall, SegFormer tends to have the top-tier performances (i.e., top or very close to top) in F1 score and IoU, demonstrating its consistency. Mask2Former also received the top F1 score, Recall, and IoU in two cases, though followed tightly by SegFormer. As expected, performance drops again notably in the single-vs-all scenarios (i.e., the "-hard" modes in the table), reflecting the difficulty of current models' generalizability in unseen conditions. However, this is a frequently encountered situation in large-scale mapping that need new developments. In comparison to Table 2, the results are slightly better, likely because in cross-region situations, there are more factors contributing to the variability, further reducing the representativeness of highly localized samples.

# 5    Conclusion and Limitations

TreeFinder offers a high-resolution, large-scale benchmark dataset for individual-level tree mortality mapping with extensive manual labels. Spanning 1,000 sites over 48 states in the CONUS with a 23,000-hectare coverage, TreeFinder supports the development of ML models capable of identifying these fine-granularity events with less contrast over different geographic regions. The dataset is enriched with metadata on climate zones and primary forest types to facilitate generalization tests under various scenarios. We consider a suite of baseline models including both convolutional and ViT-based foundational models across a wide range of generalization scenarios. Our benchmarking experiments highlight the challenges of model generalization across geographic, climate, and forest type conditions and the needs for further model developments. The dataset and corresponding Python libraries are shared to support convenient data usage.

**Limitations and future directions.**   Despite its scale and scope, TreeFinder has several limitations. First, NAIP offers near wall-to-wall coverage across CONUS, but is not available at the global scale. Future expansions may include other regions with wall-to-wall coverage of high-resolution images at national-scale (e.g., from Switzerland) or commercial satellites such as WorldView-3, which offer similar spatial resolution to NAIP over the globe, though the data may not be publicly available for free. Second, the dataset has not yet considered challenges related to the changes in NAIP dataset itself, including the change of resolution over time. Currently, we only included recent years' images at 0.6m resolution, and future extensions are needed to include 1m resolution data to support cross-resolution model development. Third, our evaluation has not considered cases for active learning, meta-learning, etc. The presented scenarios are most commonly encountered situations in practical applications, but future extensions should develop standard testing cases for different types of training strategies as well. We may also explore the applications of emerging general-purpose vision foundation models (e.g., SAM2 and DINOv2) for this challenging segmentation task. Finally, TreeFinder has not considered integration of multiple data sources (e.g., NAIP in combination with other lower-resolution platforms with richer multispectral infomration).

# Acknowledgments

Zhihao Wang, Yiqun Xie, Ruichen Wang, and Zhili Li are supported in part by the NSF under Grant No. 2126474, 2147195, 2425844, and 2530610; NASA under grant 80NSSC25K0013 and 80NSSC25K7221; Google's AI for Social Good Impact Scholars program; and the Zaratan cluster at the University of Maryland. George Hurtt is supported by the NASA CMS under Grant No. 80NSSC25K7221 and NASA EIS under Grant No. 80NSSC22K1733. Lei Ma is supported by the NASA ECIPES under Grant No. 80NSSC24K1632. Xiaowei Jia was partially supported by the NSF under Grant No. 2239175, 2147195, 2316305, 2425845, 2530609, 2203581; NASA under Grant No. 80NSSC24K1061 and 80NSSC25K0013; the USGS awards G21AC10564 and G22AC00266; and Pitt Momentum Funds and CRC at the University of Pittsburgh. Gengchen Mai is supported by the NSF under Grant No. 2521631. We would like to acknowledge high-performance computing support from the Derecho system (doi:10.5065/qx9a-pg09) provided by the NSF National Center for Atmospheric Research (NCAR), sponsored by the National Science Foundation.

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

# Appendix

## A  Dataset Generation

### A.1  Annotation Details

The labeling of dead trees was performed using the Google Earth Engine (GEE) platform with NAIP imagery (0.6m resolution). Imagery for each state was loaded using GEE scripts that filtered the NAIP collection by date and the state boundary. Visualization layers were configured with both true color (RGB) and false color (NIR–R–G) composites to aid visual interpretation. The false color combination enhances the spectral contrast between healthy and dead vegetation by emphasizing near-infrared reflectance differences.

The tree mortality labeling followed a strict and consistent set of criteria to ensure consistency and accuracy. For visualization, annotators first inspected a false-color composite to detect potential dead trees and then double-checked the observations using a true-color image. In terms of crown condition, a tree was considered for labeling only if at least half of its crown appeared visibly dead, indicated by discoloration toward the crown's edge or trunk, fuzziness, or exposed branches. The certainty threshold required that a tree appear clearly dead in both visualization modes before a label was assigned. To be more consistent in labeling and have higher certainty, annotators avoided over-labeling for trees that were brown but did not have structural indicators of mortality. Additionally, discoloration or standing branches that might have been attributable to seasonal senescence rather than mortality were further investigated through historical imagery to verify the tree's living status before a label was finalized.

Annotations were drawn manually using GEE's geometry tools, with one polygon per dead crown (or in some cases, connected dead patch). Each polygon contained at least ten pixels to maintain consistency and avoid mislabeling tiny ambiguous regions. Polygons were stored as feature collections in GEE, rasterized to binary masks (dead = 1, background = 0), and exported with the corresponding NAIP tiles. A custom function handled rasterization and reprojection at a spatial resolution of 0.6m.

### A.2  Validation Details

We validated our annotations following a standard protocol. Specifically, a stratified random sampling approach was used for validation sample collection, with sample size proportional to the total labeled area in each state. Each annotator provided 20% random samples from their labeled tiles for validation. Both commission (false positives) and omission (false negatives) were counted as disagreements between the annotator and validator. The final count of disaggreements were recorded for a consensus review. During the consensus review, each validator presented each disagreement and provided supporting visual evidence, while annotators were given the opportunity to explain their interpretation, including showing the historical images if necessary. All participants then voted on whether each disputed sample should be retained or removed from the final dataset. The resulting dataset represents a majority-voted consensus designed to minimize individual bias and ensure consistent labeling quality. The final agreement score was calculated as the number of correctly assigned labels divided by the total number of labels, leading to about 97% cross-annotator agreement.

## B  Training details

All benchmark models were trained or fine-tuned (when pretrained weights were available) using the TreeFinder dataset. The training of all models were performed with a batch size of 32 and a maximum of 100 epochs with early stopping. We used a composite loss function combining Binary Cross-Entropy (BCE) loss and Dice loss to address class imbalance and improve segmentation performance. Although we experimented with Focal Loss, commonly used for imbalanced classification, we found it yielded similar performance to BCE and therefore did not include it in the final benchmarks. All models were optimized using the AdamW optimizer with a weight decay of 0.01 and an initial learning rate of $1 \times 10^{-4}$, decayed over time using an ExponentialLR scheduler. Early stopping was applied based on validation loss with a patience of 5 epochs to prevent overfitting. Fig. 5 shows the loss changes in training and validation dataset for the random split experiment, where 80% of the dataset is used for training and 10% of the training set is reserved for validation. Both training and

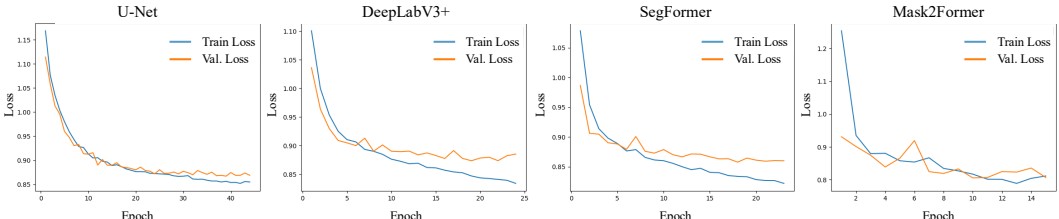

Figure 5: Training and validation loss curves for the random split experiment, where 80% of the dataset is used for training and 10% of the training set is reserved for validation.

Table 4: Performance across shifted domains in climate zones. The climate-swap scenario is added as a complementary part of Table 3 in the main paper. All values are percentages %, and best values are bolded.

| Model | Scenario | F1 | | Precision | | Recall | | IoU | | Accuracy |
|---|---|---|---|---|---|---|---|---|---|---|
| | | Dead Tree | All | Dead Tree | All | Dead Tree | All | Dead Tree | All | |
| U-Net | Climate | 18.6 | 59.1 | 17.3 | 58.5 | 20.1 | 59.8 | 10.3 | 54.8 | 99.3 |
| DeepLabV3+ | Climate | 25.8 | 62.8 | 31.4 | 65.5 | 21.9 | 60.9 | 14.8 | 57.2 | **99.5** |
| ViT | Climate | 21.6 | 60.7 | 23.4 | 61.5 | 20.1 | 59.9 | 12.1 | 55.8 | 99.4 |
| SegFormer | Climate | 28.0 | 63.9 | **33.7** | **66.7** | 24.0 | 61.9 | 16.3 | 57.9 | **99.5** |
| Mask2Former | Climate | **29.8** | **64.7** | 25.0 | 62.4 | **36.8** | **68.2** | **17.5** | **58.4** | 99.3 |
| DOFA | Climate | 12.9 | 56.3 | 15.7 | 57.6 | 10.9 | 55.3 | 6.9 | 53.1 | 99.4 |
| U-Net | Climate-swap | 29.9 | 64.9 | 44.9 | 72.3 | 22.4 | 61.2 | 17.6 | 58.6 | **99.7** |
| DeepLabV3+ | Climate-swap | 31.2 | 65.5 | 42.7 | 71.2 | 24.6 | 62.2 | 18.5 | 59.1 | **99.7** |
| ViT | Climate-swap | 23.8 | 61.8 | 30.1 | 64.9 | 19.7 | 59.8 | 13.5 | 56.6 | 99.6 |
| SegFormer | Climate-swap | 34.9 | 67.4 | **46.5** | **73.1** | 28.0 | 63.9 | 21.1 | 60.4 | **99.7** |
| Mask2Former | Climate-swap | **35.5** | **67.7** | 45.6 | 72.7 | **29.0** | **64.5** | **21.6** | **60.6** | **99.7** |
| DOFA | Climate-swap | 17.9 | 58.9 | 22.5 | 61.1 | 14.9 | 57.4 | 9.9 | 54.7 | 99.6 |
| U-Net | Climate-hard | 19.9 | 59.8 | 26.2 | 62.9 | 16.1 | 58.0 | 11.1 | 55.3 | 99.5 |
| DeepLabV3+ | Climate-hard | 18.0 | 58.9 | 36.2 | 67.9 | 12.0 | 56.0 | 9.9 | 54.7 | **99.6** |
| ViT | Climate-hard | 11.4 | 55.6 | 22.7 | 61.2 | 7.6 | 53.7 | 6.0 | 52.8 | 99.5 |
| SegFormer | Climate-hard | **22.6** | **61.2** | 33.1 | 66.4 | **17.1** | **58.5** | **12.7** | **56.1** | 99.5 |
| Mask2Former | Climate-hard | 15.4 | 57.6 | **45.8** | **72.7** | 9.2 | 54.6 | 8.3 | 54.0 | **99.6** |
| DOFA | Climate-hard | 4.7 | 52.1 | 3.9 | 51.8 | 6.0 | 52.7 | 2.4 | 50.7 | 99.0 |

validation curves gradually decrease and converge without significant divergence, indicating no clear overfitting. For each model, the checkpoint achieving the lowest validation loss was selected for final testing.

## C  Additional Results

**Cross-climate and cross-forest-type generalizability.**    This appendix provides additional results in Tables 4 and 5 to show a more complete evaluation of model generalization under shifted domains in both climate zones and primary tree types. Specifically, in this main paper we provided the results for the "Climate" and "Forest" scenarios and skipped the train-test swapped versions that were shown for the cross-region generalization test (i.e., "W-E" as a swapped version for "E-W"). Here we provide the full results where in the "Climate-swap" scenario the data in the training climate zones of "Climate" are used as testing and those in the testing climate zones are used for training. The same swapping is done for "Forest-swap" as well where forest types are used instead of climate zones. We did not do the swapping for the hard scenarios, i.e., single state as training in cross-region generalization ("CO"), single climate zone as training in cross-climate generalization ("Climate-hard"), and single forest type as training in the cross-forest-type generalization ("Forest-hard"), because their evaluation goal is to use limited samples for training and see how the model behaves in the more challenge situations. Thus, swapping them will not no longer serve this specific purpose. Looking at the results, we observe the similar performance drops relative to random splits for both scenarios, consistent with expectations due to cross-climate and cross-forest-type variability. SegFormer and Mask2Former consistently rank among the top-performing models across most metrics. In the climate-swap scenario, SegFormer achieves the highest precision, while Mask2Former achieves the highest recall, F1 score, and IoU. In the forest-swap scenario, Mask2Former outperform the other models, tightly followed by SegFormer.

Table 5: Performance across shifted domains in primary tree types. The forest-swap scenario is added as a complementary part of Table 3 in the main paper. All values are percentages %, and best values are bolded.

| Model | Scenario | F1 | | Precision | | Recall | | IoU | | Accuracy |
|---|---|---|---|---|---|---|---|---|---|---|
| | | Dead Tree | All | Dead Tree | All | Dead Tree | All | Dead Tree | All | |
| U-Net | Forest | 35.3 | 67.5 | 34.3 | 67.0 | 36.3 | 68.0 | 21.4 | 60.4 | 99.5 |
| DeepLabV3+ | Forest | 36.1 | 67.9 | 36.9 | 68.3 | 35.3 | 67.5 | 22.0 | 60.8 | 99.5 |
| ViT | Forest | 26.5 | 63.1 | 27.3 | 63.5 | 25.8 | 62.8 | 15.3 | 57.4 | 99.4 |
| SegFormer | Forest | **40.1** | **69.9** | **41.5** | **70.7** | 38.7 | 69.2 | **25.1** | **62.3** | **99.5** |
| Mask2Former | Forest | 38.6 | 69.2 | 36.8 | 68.3 | **40.6** | **70.1** | 23.9 | 61.7 | 99.5 |
| DOFA | Forest | 13.9 | 56.8 | 14.4 | 57.0 | 13.3 | 56.5 | 7.5 | 53.4 | 99.4 |
| U-Net | Forest-swap | 28.9 | 64.4 | 36.1 | 67.9 | 24.1 | 62.0 | 16.9 | 58.2 | **99.6** |
| DeepLabV3+ | Forest-swap | 26.5 | 63.1 | 35.4 | 67.6 | 21.2 | 60.5 | 15.3 | 57.4 | **99.6** |
| ViT | Forest-swap | 10.7 | 55.2 | 13.9 | 56.8 | 8.7 | 54.3 | 5.7 | 52.6 | 99.5 |
| SegFormer | Forest-swap | 32.1 | 66.0 | 39.7 | 69.7 | 27.0 | 63.4 | 19.1 | 59.4 | **99.6** |
| Mask2Former | Forest-swap | **33.8** | **66.8** | **44.1** | **71.9** | **27.4** | **63.6** | **20.3** | **60.0** | **99.6** |
| DOFA | Forest-swap | 14.9 | 57.3 | 15.1 | 57.4 | 14.6 | 57.1 | 8.0 | 53.7 | 99.4 |
| U-Net | Forest-hard | 14.7 | 57.0 | 9.6 | 54.7 | 31.5 | 65.3 | 7.9 | 53.3 | 98.8 |
| DeepLabV3+ | Forest-hard | 21.5 | 60.7 | 27.5 | 63.6 | 17.7 | 58.8 | 12.1 | 55.8 | **99.6** |
| ViT | Forest-hard | 19.9 | 59.7 | 14.5 | 57.1 | 31.7 | 65.5 | 11.1 | 55.1 | 99.1 |
| SegFormer | Forest-hard | **37.7** | **68.7** | **38.3** | **69.0** | 37.1 | 68.5 | **23.2** | **61.4** | **99.6** |
| Mask2Former | Forest-hard | 11.1 | 55.4 | 10.4 | 55.0 | 11.9 | 55.8 | 5.9 | 52.6 | 99.4 |
| DOFA | Forest-hard | 14.3 | 56.9 | 11.2 | 55.4 | 20.0 | 59.7 | 7.7 | 53.4 | 99.1 |

DOFA did not perform very well in both scenarios, likely due to its focus on cross-wavelength applicability and limited specialization for specific bands and tasks. The overall patterns are similar to those from the main paper's results.

