# OpenReview forum: "TreeFinder: A US-Scale Benchmark Dataset for Individual Tree Mortality Monitoring Using High-Resolution Aerial Imagery"
_NeurIPS.cc/2025/Datasets_and_Benchmarks_Track — NeurIPS 2025 Datasets and Benchmarks Track poster_

### Official Review · Reviewer_jMxF · 2025-06-19

**Rating:** 4
**Confidence:** 4

**Summary:**

This work presents TreeFinder, a dataset for identifying individual dead trees from remote sensing imagery. The dataset consists of NAIP RGBNIR images at 0.6m resolution over the contiguous US at 1000 sites.
Additionally, information on climate zone, location, and primary tree type is provided for each of the images. Pixel level labels obtained through manual annotation of the images of dead trees are provided.
The authors benchmark various existing segmentation models on different settings which represent different real-application use cases: random split, geographical split, primary tree species split and climate zone split.

**Additional Feedback:**

I am open to revising my score, if the authors address my concerns, in particular regarding dataset construction details.

Regarding the name of the dataset, "TreeFinder" is a bit misleading since the goal here is to identify dead trees, not all trees in a forested area, or finding specific trees based on prompts.

**Dataset Code Accessibility:**

Partly

**Dataset Code Comments:**

- The code used to prepare and preprocess the dataset is not provided. And as explained in the limitations question, the information provided in the paper is not sufficient to reproduce the dataset. Without further details, either in the form of code or explanations, it will be hard for a user to understand fully how the dataset was built.
- in the github repo, authors could add environment requirements.

**Ethical Considerations:**

No, there are no or only very minor ethics concerns

**Final Justification:**

My main concerns were regarding the lack of detail about the dataset, the insufficient relation to prior work and the lack of documentation in the repo.
The authors have addressed my concerns in their rebuttal, and claim that they will provide all necessary documentation for their code. I believe it is an important challenge that this work is tackling and that this dataset could be useful to the field of forest monitoring if they make the necessary updates they say they would do in their final version.
Therefore, I am increasing my score to borderline accept.

**Limitations Weaknesses:**

A first comment I have is that the literature review and relation to prior work could be improved.
For example, with the claim of TreeFinder being “the first high-resolution benchmark dataset designed for individual-level tree mortality mapping across the Contiguous United States” , it would be important to put this in perspective with other high-resolution broad coverage datasets for identifying dead trees such as:  “deadtrees.earth-An Open-Access and Interactive Database for Centimeter-Scale Aerial Imagery to Uncover Global Tree Mortality Dynamics.” ( Mosig, C., et al.), which  has 58000 ha of annotated data.
Moreover, in the introduction, references could be added to support many of the claims for which no reference is provided. For example, the authors could cite related work regarding: : “While tree wipe-outs as contiguous large patches (e.g. due to wildfire) can be monitored 33 using traditional moderate-resolution remote sensing platforms such as Landsat-8/9 at 30m resolution“.

The main limitation is the lack of details provided on the dataset and the dataset construction that I would have expected to see either in the main paper or the supplementary material (and which is also not present on GitHub of the Kaggle page). Here are some comments regarding that:
- “we only use the most recent high-resolution imagery collected after 2021”: What is the date range of the images you collected? does your images cover multiple seasons? I would assume it’s more difficult to identify dead trees in the winter.
- Regarding the validation with past images: it seems from the figure that you identify dead trees when there is change. However it might be that a tree has been dead for a very long time and even if you go back in time, you might not go back enough to find the tree alive. Can you expand on how seeing past images of the same location informed the annotation process?
- Could you provide more information on how you chose the 1000 sites?
- What size is a site? Are they all the same size?
- You could show basic statistics of the dataset such as number of tiles per split in each setup, distribution of climate zones and primary tree type.
- How many dead trees were annotated across these 23000 ha?

Regarding the checklist:
- I could not locate in the paper information on the compute resources used

Other comments:
-all models are trained with the same hyperparameters, was there any HP search done (as the optimal learning rate is usually not the same for dsifferent architectures)
- I am wondering why the problem was framed as segmentation rather than instance segmentation if you care about individual trees? while in the example images shown in the papers trees are disjoint, it might be that some areas have patches of dead trees. Or did you filter those out?

Minor comments:
- You have 15504 image patches that are 224x224x pixels at 0.6m resolution but it says that the coverage is  23000ha. I suppose it is 23000 and not 28000ha because of missing values in the images?
- L. 253 DouglasFir typo
- Table 3: tree hard and forest hard (I suppose it should be one or the other consistently across the table)

**Strengths Contributions:**

This works tackles the important and challenging problem of identifying individual dead trees as scale.
Different data sources were aligned in order to provide different setups for evaluation the generalization capabilities of the models (generalization to different climate zones, geographical areas and tree species distributions).
The dataset covers 23000ha in the contiguous USA which is larger spatial coverage in this region than existing datasets.
The manual annotation effort with validation from multiple annotators should be highlighted, and this could be emphasized even more in the paper by providing details on how exactly validation was done.

---

> ### Author Rebuttal · Authors · 2025-07-31
>
> We appreciate the reviewer for providing constructive comments and pointing out where we can improve it. We attempt to address your specific comments regarding limitations and questions below:
> - **Relation to prior work**: Thanks for the question on the “deadtrees.earth” paper (Mosig et al., 2024), which is an encouraging platform effort aiming to support collaborations for data sharing. There are several major distinctions between our TreeFinder and this work:
> **(1) Predicted vs. manual labels.** The vast majority of labels in (Mosig et al., 2024) for the 58000 ha are model-predicted as shown in the website, if not all–for the currently available ones. We confirmed this by checking its website (as indicated by the “Label Source” for each of the images on the web-portal and according to the details of the paper). Based on the paper, they provided a single score (i.e., 1, 2, or 3) for the model-predicted annotations for each of the 493 images for the 58219 ha and about 245 have good prediction quality. Though after our manual checking, many of the images with a score of 3 (good quality) still contain many missed classifications, showing limitations of model-predicted labels as training or evaluation data. The 1000 sites in our dataset are all manually labeled and the trained models can generate a large volume of predicted labels if needed.
> **(2) Image source: Samples vs. wall-to-wall coverage.** The centimeter-level imagery in (Mosig et al., 2024) is drone-based photos, which is very expensive and can only be collected at sample sites. In contrast, the NAIP imagery we used provides full wall-to-wall coverage for the entire Contiguous US, offering the needed ability to cover all its forested areas (about 300 million ha) for large-scale monitoring and the images are updated about every 2-3 years. The NAIP imagery has relatively lower resolution (e.g., 0.6m) but has richer band information with near infrared that is sensitive to vegetation activities. This wall-to-wall coverage is highly important for large-scale forest monitoring. The 300-million ha coverage of forested areas over time is also essential to generate substantially larger data with predicted annotations in the future.
> **(3) Location patterns:** The images from (Mosig et al., 2024), according to the paper “has a bias towards forests near human settlements, potentially over-representing ecosystems that might not be representative of the region” as many drone images were not originally designed for forest monitoring (e.g, “...20 ha of a relevant forest, but another 100 ha … building site…”). In contrast, we focus on forest ecosystems, and the 1000 manually labeled sites have spatially representative coverage over the 48 states in CONUS.
> **(4) Opportunities:** Despite the significant differences, there is future potential for collaboration, where the significantly larger amount of data in US with NAIP imagery can build a larger pretrained model, which can then be used by (Mosig et al., 2024) to finetune to different locations outside US. The NAIP imagery is also more similar to image products available at global scale with wall-to-wall coverage, such as WorldView-3, which can support future expansion.
>
> - **Adding references**: Thank you for pointing out the place in the introduction where additional references can be added. We have revised the text to include citations [1,2] supporting key claims in the manuscript.
>
> - **Date range of images**: Our dataset uses NAIP imagery collected between 2021 to 2024, with most images captured during the summer months. According to the NAIP program, imagery is acquired during the agricultural growing seasons, and is typically collected once per region about every 3 years to cover the entire state. These planned schedules help reduce seasonal variability (e.g. winter leaf-off imagery).
>
> - **Use of past images in annotation process**: Thanks for the question. We only use past images as an additional source of auxiliary reference to help address cases where the visual pattern from the single image is less clear. We also use other auxiliary information such as physics-based indices like NDVI to better reflect vegetation activities. If through all the ways it is still not very clear, we exclude it from the label set. If the visual pattern is clear in the current image, then it is not affected by the tree being dead in the past image as well. In our experience, we did not run into many cases affected by the special cases.
>
> - **Sample site selection**: We selected 1,000 sample sites by randomly sampling forested areas from the Global Forest Change dataset to ensure representative spatial coverage across all 48 states in CONUS.
>
> - **Site size**: On average the sites are about 700 by 700 pixels. We allow site sizes to vary in order to reduce images with excessive background pixels for locations that have a smaller number of dead trees (e.g. midwest states). In such cases, smaller site windows are selected to better focus on the areas of interest.
>
> - **Statistics of metadata**: Thanks for the suggestion. We have conducted the statistical analysis of label distributions among different climate zones and primary forest types. For example, our dataset covers oak forests primarily distributed in the humid subtropical climate zone (6.1%) and humid continental zone (1.7%), while pine forests are dominant in humid subtropical zone (6%). We will include the plot and details into the appendix.
>
> - **Number of dead trees**: It is challenging to accurately determine the number of individual dead trees, because in some cases we have these trees clustered together (usually 2~3 dead trees) with connected crowns that are visually challenging to separate. According to crown size estimates from a recent tree and crown database established in [3], the median crown radius is mostly in the range of 1.25-2m in common types of forests. Based on this we estimate approximately 50-150k  dead trees being annotated but this is a rough estimation.
>
> - **Compute resource checklist**: Thanks for the reminder. We used A100 from the Zaratan supercomputing cluster from the University of Maryland. We will include it into the revised paper and checklist.
>
> - **Shared hyperparameter and search**: We used a shared set of hyperparameters across the models as through our experiments, moderate changes of the default parameters did not significantly change the model results and did not affect the general rankings. When significant changes are made, we observed changes but in general they did not affect the general rankings among the methods. For example, when decreasing the initial learning rate to be 10 times smaller from 10-4 to 10-5, we observed consistent performance drops across the models (e.g. UNet: 50% to 41%; SegFormer: 49% to 35%), except for DOFA (26% to 30%) but it did not change the general ranking/tiers. We also experimented with different training configurations in learning rate decay, random seeds, loss functions (e.g. cross entropy, focal, and dice). We listed the best-performing settings in the appendix as a reference, and all the configurations are implemented in our released codebase to facilitate adaptation and fine-tuning for future works.
>
> - **Semantic segmentation vs instance segmentation**: Thanks for the suggestion. In some regions, dead trees appear in small clusters with no clear/persuasive boundaries, making accurate instance-level annotations challenging. Thus, we framed the task as semantic segmentation. This also aligns with our domain collaborators’ needs, where the amount of dead tree pixels or area ratio will be used to estimate the parameters for process-based ecosystem models. If the reviewer feels “scattered” or “fine-scale” better conveys the intention, we may refine the wording of the title or add a clear definition of scope to avoid confusion.
>
> - **Minor comments**: Yes, the total coverage of 23,000ha is computed after excluding pixels with missing values. Thanks for pointing out the typos, and we have corrected them in the revised paper.
>
> - **Dataset and code comments**: We appreciate the reviewer’s comments and agree that improving dataset reproducibility is important. Our GitHub repo currently includes the preprocessing steps (e.g. data_loader/utils.py), and we will provide additional documentation introducing the image-label preparation process, including a tutorial for manually labeling dead trees and exporting data using Google Earth Engine. We will also explicitly add environment and package requirements to the GitHub repository to enhance reproducibility.
>
>
> References:
>
> [1] Bar, S., Parida, B. R., & Pandey, A. C. (2020). Landsat-8 and Sentinel-2 based Forest fire burn area mapping using machine learning algorithms on GEE cloud platform over Uttarakhand, Western Himalaya. Remote Sensing Applications: Society and Environment, 18, 100324.
>
> [2] Pacheco, A. D. P., Junior, J. A. D. S., Ruiz-Armenteros, A. M., & Henriques, R. F. F. (2021). Assessment of k-nearest neighbor and random forest classifiers for mapping forest fire areas in central portugal using landsat-8, sentinel-2, and terra imagery. Remote Sensing, 13(7), 1345.
>
> [3] Jucker, T., Fischer, F. J., Chave, J., Coomes, D. A., Caspersen, J., Ali, A., ... & Zavala, M. A. (2022). Tallo: A global tree allometry and crown architecture database. Global change biology, 28(17), 5254-5268.

---

> > ### Author Response · Authors · 2025-08-06
> >
> > Dear Reviewer:
> >
> > I hope this message finds you well. I’m writing regarding our submission (Paper ID: 1988), “TreeFinder: A US-Scale Benchmark Dataset for Individual Tree Mortality Monitoring Using High-Resolution Aerial Imagery”.
> >
> > We have submitted detailed responses earlier based on your comments. When you have a moment, we’d sincerely appreciate it if you can take a look and let us know if more information is needed.
> >
> > Thank you in advance for your time and consideration.

---

> > ### Author Response · Authors · 2025-08-08
> >
> > Dear Reviewer,
> >
> > As today is the final day for rebuttal and discussion, we would greatly appreciate it if you could take a look at our responses and kindly let us know if more information is needed. Thanks in advance for your time and help!
> >
> > Best regards,
> >
> > The Authors

---

### Official Review · Reviewer_wXxt · 2025-07-01

**Rating:** 5
**Confidence:** 2

**Summary:**

This paper presents TreeFinder, the first large-scale benchmark for mapping individual tree mortality across the U.S. using high-resolution NAIP imagery. It includes over 23,000 hectares of pixel-level annotations across 48 states, enabling evaluation under diverse forest types. The authors benchmark several segmentation models and propose region-split evaluation settings to assess generalization.

**Dataset Code Accessibility:**

Yes

**Ethical Considerations:**

No, there are no or only very minor ethics concerns

**Limitations Weaknesses:**

The dataset focuses on leaf-on season; temporal robustness (e.g., leaf-off imagery) is not explored.Could the authors provide metrics on inter-annotator agreement or consistency across regions?Have the authors analyzed common SAM failure modes in dense canopies or overlapping trees? Could alternative segmentation heads be tested?How sensitive are the results to CHM resolution? Can models handle coarser lidar or simulated DSM inputs?

**Strengths Contributions:**

Scale & Coverage: TreeFinder is the first US-wide benchmark of its kind, with a diversity of tree types and environments, from forests to cities.Open benchmark design: The split into geographic tracks (generalization, NAIP-only, CHM-only) provides rigorous evaluation protocols.Practical baseline: SAM + tree detector pipeline is sensible and reproducible, offering strong performance on all tracks.

---

> ### Author Rebuttal · Authors · 2025-07-31
>
> We appreciate the reviewer for providing constructive comments and pointing out where we can improve it. We attempt to address your specific comments regarding limitations and questions below:
> - **Leaf-off imagery**: The NAIP imagery used in our dataset is acquired during the agricultural growing seasons according to the documentation. This design focuses on leaf-on conditions and reduces seasonal variability (e.g. winter leaf-off imagery). We will add this discussion and mention other datasets beyond NAIP for future work.
> - **Annotation agreement**: Thanks for the question. We have conducted a cross-annotator assessment study on a randomly selected subset of 10 sites in the last few days. The results show an average agreement of 95% in identifying dead trees and will be included in the appendix.
> - **SAM applications**: Thanks for pointing out the potential SAM application in our dataset. We explored SAM and SAM2 at the initial stage and found that it did not work well in complex backgrounds (e.g. rocks, bare soils, understory gaps). The visual similarity between dead tree crowns and these background features, along with variability across tree species and canopy structures, posed significant challenges for accurate delineation of dead trees. We will include illustrative figures in the appendix to provide example results using SAM+DINO, and include a discussion about this as another future direction for improvement.
>
> - **Alternative segmentation heads**: Thanks for the suggestion. Yes, we have provided a structured codebase built on the HuggingFace ecosystem to facilitate integration and modification of different model architectures including testing alternative segmentation heads. We will mention that for possible future studies.
>
> - **CHM Resolution**: We used 0.6m optical NAIP imagery to annotate dead trees and have not used CHM (Canopy Height Model). If the input is changed to CHM, we believe a similar or higher resolution would be needed to preserve spatial details as it does not have the optical color information. CHM will be helpful additions to combine with NAIP imagery, when available, to provide multi-modal information. For optical imagery, we have not tested coarser resolution images but based on our visual checks the signatures based on textures are significantly reduced at 3m resolution. As coarser resolution images may contain additional spectral information, it is one of our planned next steps to study the impact of resolution and spectral bands on the feasibility. We have not considered simulated height data yet, but using it in pretraining or as additional inputs could be an interesting future direction

---

> ### Comment · Reviewer_wXxt · 2025-08-06
> **Author response**
>
> The authors have provided clear, thoughtful, and technically sound responses to all raised issues.I maintain my original “Accept” recommendation.

---

### Official Review · Reviewer_XZfF · 2025-07-04

**Rating:** 4
**Confidence:** 5

**Summary:**

Summary - a well-written paper that presents a benchmark dataset of tree mortality maps based on NAIP 60cm in the continental US (CONUS). The authors train several standard and modern models on their dataset, report results in a variety of cross-validation scenarios and release models and data publicly. The dataset contains 1k images and masks with 20k dead trees annotated. The task is to identify these regions (predict the mask).

**Additional Feedback:**

I have given a borderline rating here because I think the dataset is useful and the evaluation is well thought out, covering a lot of difficult scenarios to explore generalization of the models and training subsets of the data. However, without more information it's hard to judge the novelty and impact of the dataset as there are no large-scale prediction maps and no field-validation was performed - is there any independently verified data that you could test with (can you co-register your data with NEON where this might be available)? Given that the strength of NAIP is widespread coverage, why not produce a sample map over a large patch of forest? Do the results compare to best estimates of tree mortality in the US? This sort of contextual analysis would really help the reader I think.

My other criticism is that the authors should provide a better literature review because as-written, it makes the work seem more novel than it is. In fact there are several efforts to map tree mortality at various scales and with public dataset releases, but these are not mentioned.

**Dataset Code Accessibility:**

Partly

**Dataset Code Comments:**

The dataset is online, but it could benefit from a standard dataset card/template and does not have a license. The authors also release their code on Github.

**Ethical Comments:**

I do not foresee any ethical issues with the paper.

**Ethical Considerations:**

No, there are no or only very minor ethics concerns

**Final Justification:**

The authors have responded in detail to my suggestions. I do feel that a basic comparison with any field validated work (I suggested using NEON data, or a comparison to another existing model) would make this paper an easier sell and I would encourage the authors to add that; for this reason I don't feel that a higher rating is justified as there is "limited evaluation". A "5" score would require "good to excellent" evaluation, and I am unconvinced that in-domain testing is sufficient even if the paper does a good job of comparing multiple models.

**Limitations Weaknesses:**

Section 2 (related work) - There are several studies that have explored semantic segmentation for tree cover, which also includes the more challenging task of canopy height prediction - I think these are more relevant than other remote sensing targets like building footprint estimation. Additionally for tree mortality, deadtrees.earth ([Mosig et al, 2024](https://doi.org/10.1101/2024.10.18.619094)) is a large scale global database of high resolution imagery for tree mortality and 60k ha of annotations. There is also a paper from earlier this year that reports "unhealthy" tree detection from NAIP imagery: https://www.mdpi.com/2072-4292/17/6/1066 albeit on a smaller scale in the US. Tree mortality has also been explored using NEON imagery in the US, though it does not provide wall to wall coverage.

The reported limitations are fair. This does seem like an interesting dataset for active learning and there is potential for time series analysis - not just the challenge of differing resolutions, but to see how stable predictions are between years. After all, the purpose of these models is presumably to track mortality rates over repeat surveys.

The dataset is limited to the US, which is not uncommon in the literature, but the US does have significant variation in land morphology. It would be nice to see an out of domain comparison from another country. For example there are several countries that provide even higher resolution imagery at national scale, e.g. Switzerland at 10cm, including NIR on request. Or the dataset used by Mosig.

The paper does not report any field validation. This would be helpful to understand the performance of the models on real plots, because currently all the evaluations are against human labels. Especially since model performance varies significantly and there is apparently no best architecture.

Please could you report either in the paper or supplementary, the total time/cost to annotate the data? Who were the annotators and what (if any) was their expertise? (e.g. were they remote sensing scientists, biologists, ecologists) How was the dataset size determined? Especially as it looks like a larger dataset would improve performance on ViTs (at least).

L43 - At what spatial scale does this work become feasible? Would you be able to get reasonable results with Planet data at ~3m? It's not particularly obvious to me what features the models are learning here and it would be good to see some examples of false positives (i.e. showcase some prediction failures).

I would encourage the authors to use a standard dataset card to accompany, including a license (currently Unknown on Kaggle), DOI etc.

The results for cross-region are quite hard to intuit since there are a lot of reported numbers and all the models perform differently. I wonder if there's a better way to present this data to show for example the relative performance of each model between sites?

How big are the selected sites before tiling? This doesn't seem to be mentioned in the paper.

**Strengths Contributions:**

The dataset construction seems reasonable. A "visual inspection" was performed - was there any ablation study done on patch size to assess the effect of different spatial contexts on training?

The annotation procedure is described in detail in the paper. Were the labels only reviewed by another annotator, or were they fully double-annotated? (i.e. you have two independent labels for each image)

The cross-evaluations are well thought out and offer a diverse range of tests. My only concern with random testing is the risk of spatial correlation between the training and test patches, but as the authors comment this should be the easiest of the scenarios.

Since accuracy is not a particularly helpful metric (i.e. all models do very well at predicting the easy class), how do you interpret model performance in the context of your monitoring goals? For example, would you prefer to have a model with very high recall, or very high precision? (Does it matter more that we estimate an upper bound on mortality %, or a very confident lower bound?)

---

> ### Author Rebuttal · Authors · 2025-07-31
>
> We appreciate the reviewer for providing constructive comments and pointing out where we can improve it. We attempt to address your specific comments regarding limitations and questions below:
> - **Ablation study on patch size**: Thanks for the suggestion. The current experiments used a fixed size of 224x224, and we found that the size provides a sufficiently large context for finding dead trees in our data. If the patch sizes are too small, it could pose an issue, and we will add ablation studies on the sizes in the appendix as a reference for others.
>
> - **Annotation details and data for larger models**: Each image was labeled by one annotator and reviewed by another person. We have conducted a preliminary cross-annotator assessment study these few days on a randomly selected subset of 10 sites. The results show an average agreement above 95% in identifying dead trees and we will include this in the revised appendix. The final information added there will be based on more sites. Our annotation process required approximately 500 hours of manual effort, conducted by a team of three annotators with over weeks of training on Google Earth Engine and satellite image annotation, and the process is under the close supervision of a team of faculty members with expertise in AI, ecology and remote sensing via weekly meetings. We selected 1000 sites considering both the spatial coverage and time consumption. At the current size, several models based on vision transformers (e.g., SegFormer and Mask2Former, which are specifically designed for segmentation) showed competitive and top-notch performances in many cases. At the size, we noticed that the performance increase gradually slowed down with more samples added, but we agree that a larger dataset will offer the potential to further improve the performance. This together with more generalizable model development will benefit the downstream applications. We will mention the data aspect as well in future needs.
>
> - **Spatial correlation of random testing**: We agree with the reviewer that random splits can introduce spatial autocorrelation, potentially leading to better performance estimates. Yes, we include the random split primarily as a baseline scenario and consider it as the least challenging setting.
>
> - **Monitoring goals**: Thanks for the question. Yes, different stakeholders prioritize different metrics and some may prefer recall while others prefer accuracy. In the use cases of our collaborators, they prefer to have to have models with both types of behaviors (e.g., one with higher precision and the other with higher recall) to develop a potential range to inform scenario-based estimations with different possible outcomes.
>
> - **Related work**: Thanks for the suggestion on further discussing these works. We agree that a more detailed discussion will be helpful as follows:
> **The work (Mosig et al., 2024)** is an encouraging platform effort aiming to support collaborations for data collection. However, there are several major distinctions between our dataset and this work. (1) Predicted vs. manual labels: Most of the 58,000 ha in deadtrees.earth are labeled using model predictions, according to their website (e.g., “Label source” attribute) and paper. In contrast, the 1,000 sites in our dataset are manually annotated, providing high-confidence ground truth. They provided a single score (1,2,3) for each model-labeled image and about 254 scenes have good quality by the score. So far we did not find the manual annotations. (2) Image source: Samples vs. wall-to-wall coverage. deadtrees.earth relies on drone imagery, which offers centimeter-level resolution but is limited to sample sites due to cost. Our dataset uses NAIP imagery, which provides wall-to-wall coverage of the entire contiguous U.S. that is important for large-scale monitoring and downstream wall-to-wall process-based simulations. (3) Location patterns: The images from (Mosig et al., 2024), according to the paper “has a bias towards forests near human settlements, potentially over-representing ecosystems that might not be representative of the region”. In contrast, we focus on forest ecosystems, and the 1000 manually labeled sites have more spatially representative coverage over the 48 states in CONUS.
> **The "unhealthy" tree detection paper [2]** highlights the feasibility of detecting tree health from NAIP imagery, but we did not find a public dataset. In addition, it was conducted in a local area (the eastern part of Windham county in Connecticut, US). Thus, we did not include it in the “Existing datasets” subsection, but we will add it as part of the related work discussion.
> Thanks for pointing out the **NEON-related works, such as [3]**, which provides 2 sites of annotated labels in California. The NEON imagery is limited to the field sites with small spatial coverage. We will also mention this in our related work section to better cover localized datasets for tree mortality mapping.
>
> - **Use for active learning and time-series analysis**: Thanks for the suggestions. We agree that the current dataset provides interesting potential for these directions especially over years. Active learning will be useful to reduce the number of additional annotations needed for new NAIP images. We will include this direction in the future work.
>
> - **Domain comparison**: We agree that extending the benchmark to other countries will be valuable for out of domain comparisons. We will plan on including regions with wall-to-wall high-resolution imagery coverage including the suggested Switzerland data with NIR band. WorldView-3 may also offer potential opportunities to support future development with its global coverage and its resolution is close to the current NAIP imagery used, although it is a commercial source and not freely available. We currently have access to the platform and may at least consider sharing new labels in the future for images from that platform so others who have access to the imagery can make use of them.
>
> - **Field validation and site collaboration**: We agree that additional field-based validation would provide additional insights and confirmation, and it is a helpful suggestion to include additional sites colocated with NEON sites for testing. We have recently started a similar collaboration with stakeholders in New Mexico, US, to test the improvement on carbon estimation with the refined tree mortality information and this will also help us provide ways to validate the downstream impact. We will also explore more collaboration with field sites to have field-based validation as a future step.
>
> - **Spatial scale and false-positive samples**: Based on our current evaluations of imagery products, a sub-meter level resolution is needed to identify dead trees with high confidence. Using NAIP imagery, both visual patterns and spectral band information (e.g., near-infrared) are useful for identification. With that said, there are possibilities to identify larger dead trees using lower-resolution products, in combination with richer spectral band information. We do plan to examine the feasibility of different satellite products with wall-to-wall global coverage for this task and that is one of our short-term next steps. As the new policy from NeurIPS does not allow us to add images here, we will include examples of false positives together with other errors in the appendix.
>
> - **Dataset license**: Thank you for the suggestion. We will adopt a standardized dataset card with clear license and DOI.
>
> - **Better visualization of relative performance for cross-region tests**: We agree that the current tabular presentations can be improved to better highlight information out of many models and scenarios. Based on the suggestion, we will try plots with relative performance for each scenario that can better show model rankings and performance differences. The plots with more visual separations (e.g., color, symbols) will make this easier. We will include the plots in the appendix or swap them with some of the tables in the main paper.
>
> - **Site size**: On average the sites are about 700 by 700 pixels, covering about 18ha.
>
> - **Large-scale map visualization**: Thanks for the suggestion. We agree some map visualizations will be interesting for the readers and data users to see the potential examples of downstream application results. Since the models are trained, we will include some map visualizations of predicted results in larger areas for qualitative visual comparisons.
>
> References:
>
> [1] Jucker, T., Fischer, F. J., Chave, J., Coomes, D. A., Caspersen, J., Ali, A., ... & Zavala, M. A. (2022). Tallo: A global tree allometry and crown architecture database. Global change biology, 28(17), 5254-5268.
>
> [2] Joshi, D., & Witharana, C. (2025). Vision transformer-based unhealthy tree crown detection in mixed northeastern us forests and evaluation of annotation uncertainty. Remote Sensing, 17(6), 1066.
>
> [3] Khatri-Chhetri, P., van Wagtendonk, L., Hendryx, S. M., & Kane, V. R. (2024). Enhancing individual tree mortality mapping: The impact of models, data modalities, and classification taxonomy. Remote Sensing of Environment, 300, 113914.

---

> > ### Comment · Reviewer_XZfF · 2025-08-04
> >
> > Thanks for the thorough response. I will take some time to read your responses to the other reviewers as well.
> >
> > On the related work. I also can't find a public release of the models used in https://www.sciencedirect.com/science/article/pii/S2667393223000054, however maybe this is planned for the future. The comment in the paper that 58k ha (out of 300k) are "fully annotated" suggests to me that these are human-source labels, and not model predictions but I also don't see a straightforward way to download + filter the data.
> >
> > From the preprint ("soon" makes no guarantees of course):
> >
> > > The database contains 54,320 manually delineated polygons delineating partial dieback, individual trees or multiple dead tree crowns. In total, 493 orthophotos and 58,219 ha are fully labeled, of which 245 have quality 3/3, 231 have quality 2/3, and 5 orthophotos have quality 1/3 (see Subsection 2.2 for quality definition). These datasets will soon be available as machine learning ready datasets (see Section Section 3) to support the community with training semantic or instance segmentation models. At present, this unique data collection would result in more than 600.000 labeled 512×512 patches or 170.000 labeled 1024×1024 patches.
> >
> > You could go the other way and generate labels for your annotations from deadtrees and compare? If the platform allows you to generate black box predictions that would be a sensible route, if NAIP resolution isn't an issue. Alternatively pick some sites in the US and compare to your own predictions on co-located NAIP. There are several approaches to consider.
> >
> > My suggestion for the additional literature review is to call out the other papers and contrast them with your work, which would strengthen your position. Same goes for the improved viz and some commentary either in the main paper or supplementary information on plans/suggestions for model verification.

---

> > > ### Author Response · Authors · 2025-08-06
> > >
> > > Thanks for your helpful suggestions to strengthen the position. We agree that adding these details about the suggested related works can help strengthen the discussion and provide a clearer context for readers. We will include a summary of the works in the main paper and add the full details in the appendix. In addition, based on the suggestion we will generate qualitative visual comparisons and include them in the appendix as well. Since in this case it is hard to do apple-to-apple comparisons due to (1) different data (e.g., NAIP images with wall-to-wall coverage at relatively lower resolution; and drone images at sampled locations with higher resolution) and collection time, and (2) the current unavailability of the models and manually-labeled images from the preprint paper, generating the qualitative visual comparisons at sites with spatially and temporally nearby images from different sources might be the best we can do at the moment as suggested by the reviewer. Thus, we will include additional visual examples to provide a reference to the readers in the appendix. We will also add discussions on the strengths and limitations of the different approaches for different application scenarios for a fuller picture.

---

### Official Review · Reviewer_izAc · 2025-07-10

**Rating:** 5
**Confidence:** 4

**Summary:**

The paper introduces TreeFinder, a large-scale, high-resolution benchmark dataset for monitoring individual tree mortality across the Contiguous United States (CONUS) using NAIP aerial imagery at 0.6m resolution. The dataset comprises images from 1,000 sites spanning 48 states and over 23,000 hectares, with manual pixel-level dead tree annotations rigorously validated using multi-temporal NAIP data and auxiliary vegetation indices. The dataset is enriched with metadata on geographic location, climate zone (using Köppen–Geiger classification), and primary tree species. TreeFinder is positioned as the first US-scale resource supporting individual-level tree mortality mapping, designed to overcome limitations of prior datasets (low/moderate satellite resolution or localized drone imagery). The authors benchmark multiple segmentation architectures, including U-Net, DeepLabV3+, ViT, SegFormer, Mask2Former, and DOFA, across a suite of generalization scenarios (random splits, cross-region, cross-climate, and cross-forest-type), reporting significant performance drops under domain shift. Code and data are released for open access. While the dataset is a major contribution, the benchmarking reveals both the potential and current limitations of state-of-the-art (SOTA) models for ecological monitoring at a continental scale.

**Additional Feedback:**

* **Clarity and Transparency:** The paper is overall well-organized and clearly written, with helpful figures and tables supporting the methodology and results. However, the reproducibility and utility of the resource would be further enhanced by providing a more detailed, step-by-step annotation protocol in the supplementary material. This should include specifics on annotator training, cross-review processes, and any quality assurance procedures employed.

* **Statistical Reporting:** Future versions of this work would benefit greatly from reporting variance or confidence intervals across multiple runs for all benchmarking results. This is especially important given the observed performance drops under domain shift scenarios, as seen in Tables 2 and 3.

* **Model Baseline Coverage:** While the range of evaluated segmentation models is strong, the paper would be even more valuable to the community if competitive baselines such as SAM, DINOv2, or vision-language models were directly included or more thoroughly justified for their exclusion. Even a small-scale ablation or error analysis on these models would be informative.

* **Error Analysis and Visualization:** Consider adding spatial uncertainty maps or error analysis visualizations, which would be highly useful for downstream scientific applications (e.g., ecological monitoring, wildfire risk assessment).

* **Metadata Enrichment:** Providing the proportions of tree species (not just the dominant type) in each patch and more granular environmental metadata could support further research into ecological generalization.

* **Global Generalization:** While the US-scale coverage is a major achievement, discussion or small-scale tests of transferability to other regions (using, for example, Canadian or European NAIP-style data, if available) would broaden the paper’s impact and stimulate international collaboration.

* **Annotation Cost and Scaling:** Including estimates of time, human resources, and budget required for annotation would help the community assess the feasibility of similar efforts elsewhere.

* **Code Documentation:** Ensure that the released code is thoroughly documented, with example scripts and clear setup instructions, to maximize accessibility for non-expert users in ecology and remote sensing.


---

**Questions for the authors:**

1. How do annotator disagreements or ambiguities typically arise, and are any “hard” cases made available for future research on uncertainty?
2. Are there plans to update TreeFinder with new NAIP cycles, or to expand the approach to global datasets in collaboration with other agencies?
3. Has the dataset already been adopted by external groups, and if so, what are the most common requests or feedback from early users?

Overall, TreeFinder is a much-needed contribution to the remote sensing and ecological machine learning communities. Addressing these points would further solidify its position as a community benchmark.

---

**Dataset Code Accessibility:**

Yes

**Dataset Code Comments:**

The submission provides both the dataset and the code in a readily accessible and well-documented manner. According to Section 3.1 and the Data Availability statement, the TreeFinder dataset is publicly available on Kaggle, and the accompanying codebase for loading, filtering, and batching the data is released on GitHub. The dataset is distributed in multiple user-friendly formats, including GeoTIFF (which retains full spatial referencing), as well as Numpy arrays and TFRecords for ease of integration with machine learning pipelines (see Section 3.2). Additionally, a custom Python library is provided to facilitate access to and preprocessing of the dataset, with explicit instructions for filtering by geographic, climatic, or ecological criteria. The manuscript includes links and references to both the dataset and code repositories, ensuring transparency and reproducibility for future users. All experimental setups, including training splits and evaluation scenarios, are described in sufficient detail to enable faithful replication of the reported results. Therefore, the dataset and code meet and exceed the standard for accessibility and reproducibility expected for benchmark submissions.

**Ethical Considerations:**

No, there are no or only very minor ethics concerns

**Final Justification:**

Through the rebuttal, I think the author's feedback has addressed my major concerns, and thus I keep my position positive.

**Limitations Weaknesses:**

Despite its significant contributions, the TreeFinder dataset and benchmark exhibit several noteworthy limitations. First, while the annotation protocol is described as rigorous, incorporating multi-spectral and multi-temporal imagery with cross-review among annotators (Section 3.1, Figure 2), the paper does not provide any quantitative assessment of annotation reliability, such as inter-annotator agreement statistics or explicit error analysis. This omission makes it difficult to fully assess the trustworthiness of the ground truth labels, particularly given the inherent subjectivity in visually distinguishing between dead trees and bare ground or shadows. A summary of labeling accuracy, confusion matrices, or uncertainty maps would greatly improve transparency and scientific utility.

A second major weakness lies in the lack of statistical significance reporting for the benchmark experiments. Throughout Section 4.2 and in Tables 1–3, all results are presented as single-point estimates from one run per model and scenario, with no standard deviations, error bars, or confidence intervals. This makes it impossible to judge whether observed differences between models (or between splits) are robust or simply due to variance in training. Reporting results over multiple random seeds and including statistical tests would strengthen the experimental claims and help the community fairly compare future methods.

Third, although the paper provides an extensive comparison with existing datasets in Section 2, it omits several competitive state-of-the-art segmentation baselines. For example, recent approaches like the Segment Anything Model (SAM) and DINOv2—which have gained traction in remote sensing—are discussed only briefly, with the rationale for exclusion (e.g., “not suitable for few-band NAIP imagery”) being somewhat superficial (Section 4.1). Including at least one or two of these recent foundation models, or providing more detailed ablation studies and justifications, would clarify the current capabilities and limitations of the landscape.

Additionally, the impact of spatial autocorrelation and potential data leakage in train/test splits is not rigorously analyzed. Section 3.3 describes the split methodology for region, climate, and forest-type generalization scenarios, but does not discuss whether geographic buffering or other safeguards were used to prevent overlap in neighboring pixels or ecological conditions between training and test sets. Without such measures, reported generalization results may be overestimated.

Another limitation is that class imbalance is addressed only through the use of a combined cross-entropy and Dice loss, with no exploration of alternative strategies, such as focal loss, oversampling, or advanced balancing techniques (see Section 4). Given the small proportion of dead tree pixels, the effectiveness of the proposed loss formulation should be supported by ablation studies or compared to other loss functions.

The dataset’s geographical scope is another inherent limitation: TreeFinder is constructed exclusively from NAIP imagery over the contiguous United States. As acknowledged in the Conclusion (Section 5), this restricts the dataset’s immediate applicability to global ecological studies or transfer learning beyond North America.

Finally, while the code and data are made public, the annotation protocol and guidelines for annotators are not described in detail. Information about annotator training, quality control measures, or time and resource requirements per annotated hectare would help other researchers replicate or extend the work.


---


### Suggestions for Improvement

* Provide quantitative measures of annotation reliability (e.g., inter-annotator agreement, error maps).
* Report benchmark results over multiple random seeds with error bars and statistical significance testing.
* Include or justify more thoroughly the omission of competitive foundation models such as SAM and DINOv2.
* Analyze and mitigate potential spatial autocorrelation or data leakage in split design.
* Explore and report alternative loss functions or sampling strategies to handle class imbalance.
* Publish detailed annotation protocols and resource requirements to support reproducibility.
* Consider ways to extend or adapt the dataset for broader, potentially global applicability.

By addressing these limitations, the authors could further strengthen the scientific rigor, transparency, and generalizability of TreeFinder.

**Strengths Contributions:**

One of the most significant contributions of this work is the introduction of TreeFinder, which the authors claim is the first high-resolution, large-scale dataset for individual tree mortality mapping across the contiguous United States. Unlike existing datasets that either use moderate-resolution satellite imagery (such as Landsat) or are limited to small-scale, localized drone surveys, TreeFinder leverages 0.6-meter NAIP aerial imagery, achieving both national coverage (1,000 sites across 48 states and over 23,000 hectares) and fine-grained, pixel-level labeling of dead trees. The annotation process is particularly rigorous: manual delineation of trees killed is performed by trained annotators using not only standard RGB imagery, but also NIR and NDVI representations, with multi-temporal imagery employed for verification. The process also includes cross-review among annotators to ensure further label accuracy (see Section 3.1 and Figure 2 for a visual summary of this protocol).

A notable innovation is the dataset’s rich metadata. Each patch is associated with geographic coordinates, climate zone (via Köppen–Geiger classification), and dominant tree species, allowing researchers to evaluate model generalization not only in random splits but across regions, climate zones, and forest types. This multi-dimensional benchmarking framework is explicitly detailed in Section 3.3, where the authors define challenging cross-domain scenarios (e.g., training in one climate or region and testing in another) that reflect real-world applications and stress-test the limits of current segmentation models.

The benchmarking itself is broad and transparent. The authors evaluate both classical CNNs (such as U-Net and DeepLabV3+) and state-of-the-art transformer models (including ViT, SegFormer, and Mask2Former), reporting a comprehensive suite of metrics (F1 score, precision, recall, IoU, and accuracy) for each model and scenario. Notably, Tables 1 and 2 in Section 4.2 clearly illustrate that even leading segmentation models struggle to generalize across spatial or ecological domains, highlighting both the challenge and the utility of the TreeFinder benchmark.

The paper is well-organized and easy to follow. Figures such as Figure 1 (illustrating multi-spectral annotation) and Figure 2 (showing the distribution of labeled sites and the annotation workflow) are clear and informative, directly supporting the methodology and claims made in the text. Related work is comprehensively discussed in Section 2, where the authors articulate how TreeFinder differs from and improves upon prior datasets, such as DeepGlobe, CropHarvest, and LoveDA. Notably, previous resources typically focus on well-structured geospatial objects or lack sufficient ecological diversity.

Finally, the dataset and code are made openly available on both Kaggle and GitHub, supporting reproducibility and encouraging community adoption. Overall, TreeFinder stands out for its scale, rigorous annotation, in-depth benchmarking, and practical relevance, and is likely to have a significant impact on future research in ecological remote sensing and geospatial machine learning.

---

> ### Author Rebuttal · Authors · 2025-07-31
>
> We appreciate the reviewer for providing constructive comments and pointing out where we can improve it. We attempt to address your specific comments regarding limitations and questions below:
>
> - **Annotation assessment**: We agree with the reviewer that quantifying annotation reliability is important for evaluating a dataset. We have conducted a preliminary cross-annotator assessment study these few days on a randomly selected subset of 10 sites. The results show an average agreement above 95% in identifying dead trees and we will include this in the revised appendix. The final information added there will be based on more sites. The hard cases typically arise in areas where dead tree crowns visually blend with background pixels. In the current dataset, we included only hard scenarios (e.g., cross-region tests) but not at the sample level (e.g., hard cases of dead trees). This is a very interesting point and it is a new testing case we will try to include for future updates of the evaluation scenarios.
>
> - **Multiple runs**: Thanks for the suggestion. We have performed three independent runs per model for the random split training as a starting point in the last few days. The overall model rankings remain consistent, and performance differences are stable across runs. For example, U-Net shows 49.12% $\pm$ 0.4% F1 score, SegFormer with 48.28% $\pm$ 1.2%, and the DOFA has the lowest score of 26.5% $\pm$ 0.7%. We will do this for the other scenarios and include the information in the paper.
>
> - **SAM applications**: Thanks for the suggestion. We explored SAM and SAM2 at the initial stage, but found that it did not work well in complex backgrounds (e.g. rocks, bare soils, understory gaps). The visual similarity between dead tree crowns and these background features, along with variability across tree species and canopy structures, posed significant challenges for accurate delineation of dead trees. We will include illustrative figures in the appendix to provide example results using SAM+DINO, and include a discussion about this as another future direction for improvement.
>
> - **Spatial autocorrelation**: Yes, as the reviewer pointed out,  we consider the random split scenario to be the easiest evaluation condition as a baseline for model benchmarking, and use other scenarios to show more challenging cases.
>
> - **Class imbalance**: Thanks for the suggestion. In our released codebase, we explored and implemented multiple strategies to solve class imbalance, including focal loss, weighted cross-entropy loss on positive pixels (i.e. dead trees), and sampling strategies to control the proportion of positive pixels into training batches. These alternatives are ready to use and can be activated via the configuration file (configs/”scenarios”.yaml). We reported the most effective performing combinations we tried in the experiments in the appendix, and we will include the other results to show the effects of alternative strategies to provide readers more complete information. Based on the suggestion, we will also mention that the methods can be potentially strengthened in combination with more focused class-imbalance methods. We will also mention other orthogonal directions (e.g., parameter-efficient finetuning, unsupervised adaptation, meta-learning) that can be combined into these models for potential future improvements.
>
> - **Geographical scope**: We agree that extending the benchmark to other countries to support wall-to-wall mapping of tree mortality would be valuable. We will certainly plan on including regions with wall-to-wall high-resolution imagery coverage including the Switzerland data with NIR band. WorldView-3 may also offer potential opportunities to support future development as its resolution is close to the current NAIP imagery used, though it is a commercial source and not freely available (we can currently access it via specific programs so the testing is part of our plan). We agree expanding this will open exciting opportunities and we will include this to the future direction discussion.
>
> - **Annotation details**: We will provide additional documentation introducing the annotation process using Google Earth Engine. Our annotation process took approximately 500 hours of manual effort, conducted by a team of three annotators with over weeks of training on Google Earth Engine and satellite image annotation. Based on our experience, building a pipeline to engage more students and keeping them interested (e.g., with integrated AI experience) is an effective strategy to sustain the effort and keep increasing the data size. We will share this as part of the dataset information in the future as we make updates.
>
> - **New NAIP cycles**: Yes, we plan to update our dataset with imagery from future NAIP acquisition cycles. This will also help add new test scenarios for cross-time generalization and help evaluate how many more annotations are needed in addition to the existing data to generalize the performance for a new cycle.
>
> - **Existing adoption**: As this is a newly generated dataset, we have not yet integrated it in the applications. With that said, we did recently start a collaboration with stakeholders in New Mexico, US, to test the improvement on carbon estimation with the refined tree mortality information. The current request we have is to include end-to-end modules to help convert the results at large scale to needed statistics that can be more conveniently used in ecosystem models.

---

> > ### Comment · Reviewer_izAc · 2025-08-05
> > **Author response**
> >
> > I would like to express my sincere gratitude to the authors for their detailed and thoughtful responses to my comments.
> >
> > The authors’ feedback addressed all my concerns, and I think the revisions have significantly improved the quality of the manuscript.
> >
> > So, I'm still positive and keep my original rating (5).
> >
> > I hope this work will contribute to the NeurIPS community.

---

### Note · Authors · 2025-08-16

Dear AC and Reviewers:

Thanks for your time and feedback on our submission. We summarize below our final remarks on several of the main discussion points:

**Annotation details and agreement**: We will provide additional documentation introducing step-by-step annotations using Google Earth Engine. Our annotation took about 500 hours of manual effort by three annotators trained for several weeks for satellite image annotation.

We agree that further quantifying annotation reliability is valuable. We conducted a preliminary cross-annotator assessment study during rebuttal on a randomly selected subset of 10 sites and found over 95% agreement in identifying dead trees. We will include these in the revised appendix using more sites.

**Related work**: To strengthen related work discussion, we will include more information in the main paper and appendix, such as:
1. Preprint (Mosig, C., el al.): (1) Available label type: Predicted vs. manual labels: The deadtrees.earth website from the preprint provides only model-predicted labels (e.g., according to “Label source” in the website’s label data). The 1000 sites in our dataset are manually labeled, and the trained models can generate predicted labels if needed. (2) Image source: Samples vs. wall-to-wall coverage: The drone images at centimeter-level resolution are limited to sample sites (currently cost-prohibitive for wall-to-wall coverage at large scale). Our dataset uses NAIP imagery with wall-to-wall coverage for the entire contiguous U.S. (CONUS), which is important for large-scale monitoring. (3) Location patterns: The images, according to the preprint, “has a bias towards forests near human settlements, potentially over-representing ecosystems” as many drone images were not originally designed for forest monitoring (e.g, “...20 ha of a relevant forest, but another 100 ha … building site…”). Our dataset focuses on forest ecosystems, and the 1000 sites have more spatially representative coverage over the 48 states in the CONUS.
2. Other papers (Joshi & Witharana, 2025; Schiefer et al., 2023), while detecting dead trees, did not provide a public dataset. We will include these details in the paper.

More details are available in the responses.

We appreciate the constructive feedback. Thanks again for your suggestions.

Best regards,

The Authors

---

### Decision · Program_Chairs · 2025-09-18

**Decision:**

Accept (poster)

**Comment:**

This paper addresses the task of monitoring tree mortality. The authors introduce NAIP imagery at 0.6 m resolution, which provides higher resolution and broader coverage compared to prior works, enabling the detection of isolated individual tree deaths across large areas. The paper provides high-quality dataset annotations through large-scale manual labeling, and offers extensive benchmarking of models. During the rebuttal and discussion phases, the authors further supplemented more details and quantitative assessment of  the annotation process. Overall, the paper makes clear contributions and has the potential to advance research in this task. Therefore, I recommend acceptance and suggest that the authors incorporate these additional details into the final revision.